# LIKELIHOOD PARADOX MITIGATION USING ENTROPY MANIPULATION WITH NORMALIZING FLOW IN OOD DETECTION

## ABSTRACT

Deep generative models that can tractably obtain the likelihood of input data, such as normalizing flows, often assign unexpectedly high likelihood to out-of-distribution (OOD) inputs that were unseen during training. We address this likelihood paradox by manipulating input entropy in a way that reflects semantic similarity, so that OOD samples receive stronger perturbations than in-distribution samples. We provide a theoretical analysis that demonstrates how entropy control increases the expected log-likelihood separation toward the in-distribution, and explain why our procedure works without any additional training of the density model. We then evaluate against likelihood-based OOD detectors on standard benchmarks and find that our method consistently improves AUROC over baselines, supporting the proposed explanation.

## 1 INTRODUCTION

Out-of-distribution (OOD) detection task is considered important in fields such as manufacturing systems and medicine (Mezher & Marble, 2024; Narayanaswamy et al., 2023). A supervised, discriminative approach is possible, but it is often impractical because it assumes access to OOD data during training and coverage over diverse OOD types (Havtorn et al., 2021). To avoid this requirement, unsupervised methods based on deep generative models have been explored for OOD detection (Yu et al., 2021; Ryu et al., 2018; Ran et al., 2022). There are three representative deep generative models that can be utilized for this task: the variational autoencoder (VAE, Kingma & Welling (2013)), the generative adversarial network (GAN, Goodfellow et al. (2020)), and normalizing flow (Dinh et al., 2014). Among these models, normalizing flows are attractive because they provide tractable likelihood estimates, unlike VAE or GAN, which only provide a lower bound on the likelihood or cannot tractably calculate the likelihood (Nalisnick et al., 2018). Since flows yield tractable likelihoods, one can score test inputs by their log-likelihood; under the usual assumption that likelihood is higher for typical training data, OOD samples should receive lower values than in-distribution samples.

However, several studies have reported counterintuitive behavior in OOD detection tasks using deep generative models capable of likelihood estimation when using likelihood from flows (Nalisnick et al., 2018). For example, a model trained on CIFAR-10 (Krizhevsky et al., 2009) may assign higher likelihood to SVHN (Netzer et al., 2011) serving as the OOD dataset, than to CIFAR-10 itself (Nalisnick et al., 2018). A regularity behind this paradox was noted by Serrà et al. (2019): images with simpler textures tend to assign higher likelihoods than in-distribution images. Since compression length in bits (e.g., PNG) serves as a proxy for entropy, their complexity measure can be interpreted as an entropy estimate. Combining this term with the likelihood improves detection on CIFAR-10 (in-distribution) vs. SVHN (OOD) setting, which had previously exhibited unintuitive results. But performance can degrade or even reverse with SVHN as in-distribution and CIFAR-10 as OOD. From an analytic viewpoint, Caterini & Loaiza-Ganem (2022), decomposed the expected log-likelihood difference for in-distribution $P$, OOD $Q$, and a model $P_\theta$ as[1]:

---

[1]Without loss of generality, we assume that $D_{KL}(Q||P_\theta), D_{KL}(P||P_\theta)$ exist.

$$\mathbb{E}_{\mathbf{x}\sim P}[\log P_\theta(\mathbf{x})] - \mathbb{E}_{\mathbf{x}\sim Q}[\log P_\theta(\mathbf{x})] = D_{KL}(Q||P_\theta) - D_{KL}(P||P_\theta) + \mathbb{H}(Q) - \mathbb{H}(P) \quad (1)$$

When $\mathbb{H}(P) > \mathbb{H}(Q)$, this difference can flip signs even if $P_\theta$ closely fits $D_{KL}(P||P_\theta)$, which explains failures of raw likelihood as an OOD score. Motivated by the role of entropy in this decomposition, we manipulate test-time entropy using a pretrained feature extractor that encodes semantic information. Our method perturbs inputs with Gaussian noise whose scale depends on the maximum cosine similarity between the test embedding and a memory bank of in-distribution embeddings, and the procedure is post hoc with no additional training of the density model. We show both theoretically and empirically that increasing the entropy of the OOD by adding noise improves the separation in expected likelihoods, and that the proposed approach consistently outperforms other likelihood-based methods, thereby explaining the reasons for the success of our framework. The main contributions of our work are as follows:

- We theoretically derive entropy- and KL-divergence based lower bounds under Gaussian perturbations (Theorems 3.1 and 4.1) that specify when controlled entropy increases enlarge the expected log-likelihood separation toward the in-distribution.

- A training-free algorithm, Semantic Proportional Entropy Manipulation (SPEM), that uses a pretrained encoder and a memory bank to scale noise by semantic similarity and then performs detection with the original flow model, with a noise-only variant (SPEM-noise) isolating the effect of original image.

- Extensive evaluation across ten in/out dataset pairs, including classical failure cases, shows consistent gains over likelihood, likelihood-ratio, typicality, complexity, and recent multi-statistic baselines, together with ablations on encoder choice, ReAct, and the perturbation scale.

## 2 RELATED WORKS

### 2.1 NORMALIZING FLOW

Normalizing flows are a class of generative models, where input data $\mathbf{x} \in \mathbb{R}^d$ following distribution $P$ is transformed into $\mathbf{z} \in \mathbb{R}^d$, which typically follows a standard Gaussian distribution $\mathcal{N}(0, I_d)$ (Dinh et al., 2016), by using an invertible mapping $f : \mathbb{R}^d \to \mathbb{R}^d$ in order to estimate $P$. This generative model has the advantage that, through the change-of-variable formula, it can provide a tractable estimate of the likelihood without explicit knowledge of the true underlying distribution, which can be formally expressed as:

$$\log p_{\mathbf{x}}(\mathbf{x}) = \log p_{\mathcal{N}(0,I_d)}(\mathbf{z}) + \log \left| \det \frac{\partial \mathbf{z}}{\partial \mathbf{x}} \right|, \ \ s.t. \ \ \mathbf{x} \sim p_{\mathbf{x}}, \ \mathbf{z} \sim \mathcal{N}(0, I_d).$$

In a normalizing flow model, data generation proceeds by first sampling a latent vector $\mathbf{z}$ from the base distribution $\mathcal{N}(0, I_d)$. Then, the sampled latent vector is mapped into the data space through the inverse of the learned flow transformation $f^{-1}$.

When designing a flow architecture, it is essential that the transformation function $f$ is invertible, and that the Jacobian determinant be efficiently computable. To satisfy these requirements, one approach is to construct transformations that are both easily invertible and have tractable Jacobian determinants (Rezende & Mohamed, 2015). Another widely used strategy is to employ coupling layers, in which the Jacobian takes a triangular form, thereby enabling efficient determinant computation (Dinh et al., 2014; 2016; Kingma & Dhariwal, 2018). In addition, Behrmann et al. (2019) proposed i-ResNet, constructing flows by enforcing Lipschitz constraints on residual networks to ensure invertibility. Building on this, Chen et al. (2019) addressed the issue of biased log-density estimation inherent in i-ResNet's approach while improving memory efficiency. In addition to these methodologies, a broad range of research efforts has investigated alternative designs to enhance the flexibility and expressiveness of normalizing flow. For instance, spline-based transformations have been proposed to provide more powerful and adaptive mapping functions within normalizing flows, thereby improving density estimation performance (Durkan et al., 2019). Additionally, normalizing flows with stochastic sampling blocks relax topological constraints and achieve higher expressivity than deterministic flows (Wu et al., 2020).

## 2.2 LIKELIHOOD PARADOX OF DENSITY ESTIMATION MODELS

Nalisnick et al. (2018) demonstrated that likelihood-based density estimation models such as normalizing flows can assign likelihoods in ways that contradict human intuition. For instance, when a flow is trained on CIFAR-10 as the in-distribution and evaluated on SVHN as out-of-distribution (OOD), the resulting OOD detection performance is extremely poor, with AUROC values dropping below 10%. Building on this observation, Kirichenko et al. (2020) analyzed the underlying reasons in coupling-layer flows such as RealNVP and proposed methodological adjustments such as modifying masking strategies to alleviate the issue. Schirrmeister et al. (2020); Ren et al. (2019) improved OOD detection by training additional flow or background information and using the resulting likelihood ratio with the in-distribution model as the anomaly score. Serrà et al. (2019); Kamkari et al. (2024) proposed incorporating measures of input image complexity such as local intrinsic dimension or compression length using general-purpose compressors like PNG into the scoring function for improved OOD detection. Ahmadian et al. (2021); Morningstar et al. (2021); Osada et al. (2024) argued that relying solely on input data likelihood often leads to failures in OOD detection, and therefore proposed methods that incorporate additional statistics—such as latent likelihood, input complexity estimated via general-purpose compressors, and Jacobian determinants—within auxiliary classifiers trained to improve detection performance. Zhang et al. (2021); Le Lan & Dinh (2021) demonstrated that paradoxical behavior in likelihood assignment can arise even when a density estimation model perfectly estimates the in-distribution. Choi et al. (2018) proposed an anomaly scoring method by ensembling generative models and employing the Watanabe-Akaike Information Criterion (WAIC) (Watanabe & Opper, 2010), and Nalisnick et al. (2019) introduced an OOD detection approach based on the typicality set of in-distribution data, which can outperform a likelihood-based detection approach. Caterini & Loaiza-Ganem (2022), which forms the foundation of Sections 3 and 4 of our paper, analyzed likelihood inversion and the effectiveness of likelihood ratio–based methods from an entropic perspective.

## 3 CAN ENTROPY MANIPULATION INCREASE PERFORMANCE?

In this section, we examine whether manipulating entropy can improve the performance of likelihood-based OOD detection.

### 3.1 PROBLEM STATEMENT

We estimate the probability density function of the in-distribution $P$ and aim to learn a model $P_\theta$ (e.g., a normalizing flow) that can estimate its likelihoods using only samples $\mathbf{x} \sim P$, assuming no access to OOD data during training. After training, $P_\theta$ is used to detect OOD samples $\mathbf{y} \sim Q \neq P$ based on their likelihoods. If the likelihood assigned by $P_\theta$ to a test vector $\mathbf{x}_{\text{test}}$ is below a threshold $\alpha$, determined in-distribution, then $\mathbf{x}_{\text{test}}$ is classified as OOD. However, it has been reported that the majority of OOD data receive higher likelihood under $P_\theta$ than in-distribution data, leading to failures of likelihood-based detection. In such cases, the following inequality generally holds:

$$\mathbb{E}_{\mathbf{x}\sim P}[\log P_\theta(\mathbf{x})] < \mathbb{E}_{\mathbf{x}\sim Q}[\log P_\theta(\mathbf{x})] \tag{2}$$

### 3.2 ENTROPY MANIPULATION

Previous studies have reported that the entropy (i.e., a proxy for complexity) of input data can affect the assignment of likelihood, and unintuitive behavior often arises when $\mathbb{H}(P) > \mathbb{H}(Q)$ (Caterini & Loaiza-Ganem, 2022; Serrà et al., 2019; Osada et al., 2024). Under the decomposition in Equation 1, the expected log-likelihood difference splits into two terms: $D_{KL}(Q||P_\theta)$ and $\mathbb{H}(Q) - \mathbb{H}(P)$. Because $Q$ is unavailable during training, directly controlling $D_{KL}(Q||P_\theta)$ is infeasible. In contrast, the entropy term can be influenced by perturbing OOD samples to obtain a higher-entropy $Q'$. This insight raises the methodological question of whether manipulating entropy can systematically improve the performance of likelihood-based OOD detection:

> *If we increase OOD entropy via perturbation, wouldn't the expected log-likelihood difference align with intuition?*

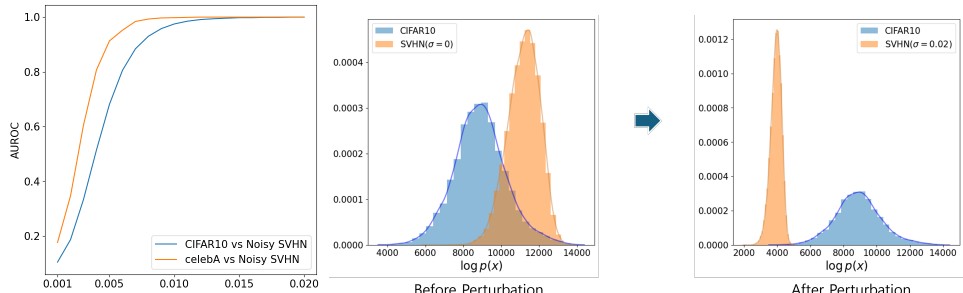

Figure 1: AUROC changes with entropy manipulation intensity and log-likelihood assignments with and without perturbation. We increase $\sigma$ from 0.001 to 0.02 and add $\mathbf{z} \sim N(0, \sigma^2 I_d)$ perturbation to SVHN to create a noisy SVHN distribution, and perform OOD detection through likelihood with Glow trained on CIFAR-10 and CelebA. The histogram shows the change in log-likelihood assignment before and after perturbation when Glow is trained with CIFAR-10.

Let $P$ be a continuous distribution on $\mathbb{R}^d$. For $X \sim P$, consider applying a Gaussian perturbation $Z \sim \mathcal{N}(0, \sigma^2 I_d)$. Then, $X + Z$ will follow the distribution of $P'$ obtained by convolution of $P$ and $Z$, which intuitively satisfies $\mathbb{H}(P') > \mathbb{H}(P)$ due to uncertainty in perturbation (A formal proof appears in Appendix A for Theorem 3.1). Hence, we may be able to mitigate the likelihood inversion by increasing the entropy term of OOD that contributes to the expected likelihood difference.

To verify this hypothesis, we examine how detection performance changes as OOD entropy increases. We adopt settings known to exhibit likelihood inversion: CIFAR-10 or CelebA (Liu et al., 2015) image datasets as the in-distribution and SVHN as the OOD dataset, training Glow (Kingma & Dhariwal, 2018) on each in-distribution dataset. The in/out-of-distribution pairs used in our experiments are those for which likelihood inversion has been reported in density estimation models, making them well-suited for testing our hypothesis (Kirichenko et al., 2020; Kamkari et al., 2024). We then added perturbations $Z \sim \mathcal{N}(0, \sigma^2 I_d)$ to SVHN samples and examined how OOD detection performance changes with different scales of $\sigma^2$. The corresponding experiment is shown in Figure 1, and the following observations were obtained.

**Observation 1.** *When detection fails because the OOD has lower entropy than the in-distribution, AUROC increases as the Gaussian perturbation scale applied to the OOD grows.*

Figure 1 demonstrates that likelihood-based OOD detection yields low performance for very small $\sigma$, but the AUROC rapidly converges to 1 when $\sigma$ is greater than about 0.01. The histograms further indicate that perturbed log-likelihoods move in the intuitive direction at the sample level. These observations provide empirical evidence that increasing OOD entropy via perturbation can enhance likelihood-based detection. To explain why this improvement can occur, we derive Theorem 3.1 by extending Equation 1, and a formal proof is provided in Appendix A.

**Theorem 3.1.** *Let $P$, $P_\theta$, $Q$, be $d$-dimensional continuous probability distributions on $\mathbb{R}^d$. Let $X \sim Q$, $Z \sim \mathcal{N}(0, \sigma^2 I_d)$, and define $Q'$ as the distribution of $X + Z$. Then a lower bound on the expected log-likelihood difference estimated by $P_\theta$ between $P$ and $Q'$ is*

$$\frac{d}{2} \log \left( e^{\frac{2}{d} \mathbb{H}(X)} + 2\pi e \sigma^2 \right) - \mathbb{H}(P) - D_{KL}(P || P_\theta).$$

By Theorem 3.1, the lower bound increases with the perturbation scale $\sigma$. This suggests that increasing the entropy gap, $\mathbb{H}(X + Z) - \mathbb{H}(P)$, helps the expected log-likelihood difference to align with our intuitive ordering. Note that interpreting performance variations purely as a function of the intensity of semantic information degradation is misleading; prior work shows that applying entropy-reducing transformations to the original distribution $Q$, such as collapsing inputs to a constant image, can worsen likelihood inversion (Osada et al., 2024).

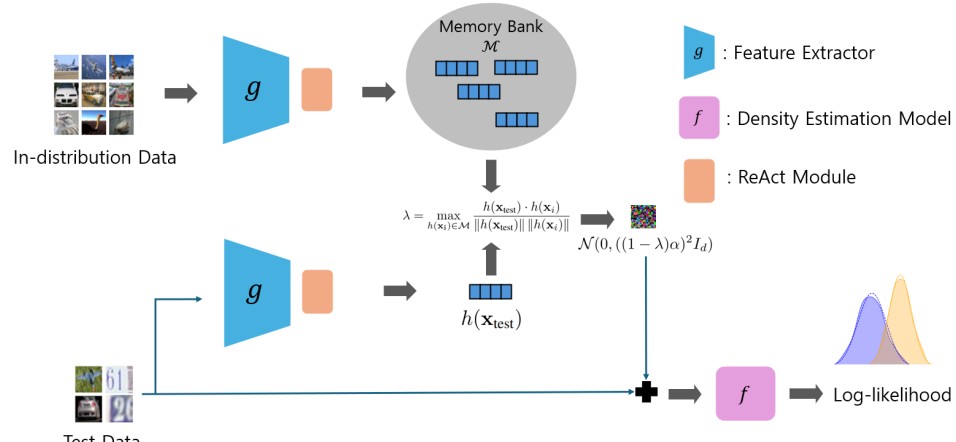

Figure 2: The overall framework of SPEM. $f$ is a density estimation model that provides a tractable likelihood for the input vector and estimates the in-distribution, and $g$ is a feature extractor pretrained with general image data, capable of sufficiently extracting features for each image.

## 4 SEMANTIC PROPORTIONAL ENTROPY MANIPULATION

In the previous section, we demonstrated that manipulating entropy through perturbations of the OOD data can improve OOD detection performance. However, at inference time we do not know whether a test sample comes from the in-distribution or from OOD, so selectively perturbing only OOD samples is infeasible. This raises a practical question:

> *Can we improve detection by increasing entropy more for OOD-like inputs than for in-distribution-like inputs?*

To analyze the effect of perturbations, we focus on the term $\mathbb{H}(Q) - \mathbb{H}(P) - D_{KL}(P||P_\theta)$, which forms part of the likelihood expectation difference under the setting $X \sim P$, $Y \sim Q$, and $P_\theta$ trained to estimate $P$. Assume that the KL-divergences among $P$, $Q$, and $P_\theta$ exist. Suppose we apply a weak perturbation $Z$ to $P$ and a stronger perturbation $Z'$ to $Q$, yielding perturbed distributions $X + Z \sim P'$ and $Y + Z' \sim Q'$. Under this setting, the lower bound for the gain in the likelihood expectation difference before and after applying perturbations to the in-distribution and OOD can be derived as follows:

$$
\begin{aligned}
& \mathbb{E}_{\mathbf{x}\sim P'}[\log P_\theta(\mathbf{x})] - \mathbb{E}_{\mathbf{x}\sim Q'}[\log P_\theta(\mathbf{x})] - (\mathbb{E}_{\mathbf{x}\sim P}[\log P_\theta(\mathbf{x})] - \mathbb{E}_{\mathbf{x}\sim Q}[\log P_\theta(\mathbf{x})]) \\
& = D_{KL}(Q'||P_\theta) - D_{KL}(P'||P_\theta) + \mathbb{H}(Q') - \mathbb{H}(P') \\
& \quad - (D_{KL}(Q||P_\theta) - D_{KL}(P||P_\theta) + \mathbb{H}(Q) - \mathbb{H}(P)) \\
& \geq (\mathbb{H}(Q') - \mathbb{H}(Q)) - (\mathbb{H}(P') - \mathbb{H}(P) + D_{KL}(P'||P_\theta) - D_{KL}(P||P_\theta)) - D_{KL}(Q||P_\theta).
\end{aligned}
\tag{3}
$$

According to Equation 3, if the entropy increase for the in-distribution, together with the corresponding KL-divergence increment, $\mathbb{H}(P') - \mathbb{H}(P) + D_{KL}(P'||P_\theta) - D_{KL}(P||P_\theta)$, is smaller than the entropy increase for the OOD, $\mathbb{H}(Q') - \mathbb{H}(Q)$, then the lower bound of the gain in the likelihood expectation difference increases. The remaining issue is how to assign different perturbation strengths to in-distribution and OOD samples. To this end, we propose **Semantic Proportional Entropy Manipulation (SPEM)**, a methodology that selectively adjusts perturbation intensity per sample using semantic similarity so that OOD-like inputs receive stronger perturbations than in-distribution-like inputs. This design manipulates OOD entropy efficiently without additional training, providing a lightweight mechanism to enhance likelihood-based OOD detection.

The proposed algorithm proceeds in three steps: first, given the in-distribution dataset

$$
\mathcal{X}_{\text{in}} = \{\mathbf{x}_i\}_{i=1}^n,
$$

we train a density estimation model $f$, such as a normalizing flow, that provides tractable likelihoods for input data. Next, we construct a memory bank $\mathcal{M}$

$$\mathcal{M} = \{h(\mathbf{x}_i)\}_{i=1}^n, \ \ s.t. \ h(\mathbf{x}_i) = \{\min(g(\mathbf{x}_i)_j, \beta)\}_{j=1}^{d'}$$

where $g : \mathbb{R}^d \to \mathbb{R}^{d'}$ is a feature extractor pretrained on general image data like ImageNet (Deng et al., 2009), and $g(\mathbf{x}_i)_j$ denotes the $j$-th element of $g(\mathbf{x}_i)$. Each element of $\mathcal{M}$ stores the embedding vector of an in-distribution sample. Additionally, we apply ReAct (Sun et al., 2021), which mitigates OOD overconfidence in the feature extractor, by rectifying the activation, which can be expressed as $h(\mathbf{x}_i)$ and $\beta$ is the limit of maximum activation determined by $p$-th quantile of activations. Through this process, the memory bank $\mathcal{M}$ stores compressed $d'$-dimensional embeddings rather than raw images, capturing semantic information for more expressive representations. Finally, for a test vector $\mathbf{x}_{\text{test}}$, we manipulate its entropy by applying the following Gaussian perturbation:

$$\mathbf{x}'_{\text{test}} = \mathbf{x}_{\text{test}} + \mathcal{N}(0, ((1-\lambda)\alpha)^2 I_d), \ \ s.t. \ \lambda = \max_{h(\mathbf{x_i}) \in \mathcal{M}} \frac{h(\mathbf{x}_{\text{test}}) \cdot h(\mathbf{x}_i)}{\|h(\mathbf{x}_{\text{test}})\| \, \|h(\mathbf{x}_i)\|}$$

where the perturbation intensity $\lambda$ is adaptively scaled according to the maximum cosine similarity between the test vector and the stored in-distribution embeddings and $\alpha$ is a hyperparameter that controls the overall strength of entropy manipulation. Since the in-distribution is assumed to contain multiple heterogeneous classes, we employ the maximum cosine similarity rather than the mean. Finally, we define the anomaly score as the log-likelihood estimated by the density estimation model $f$, and perform detection by classifying samples with lower likelihood values as OOD.

SPEM enables likelihood-based OOD detection to remain effective under both possible entropy orderings between in-distribution and OOD, whereas existing methods are typically unable to perform reliably across both cases (Osada et al., 2024). When the in-distribution exhibits higher entropy than the OOD, the similarity measure $\lambda$, defined as the maximum cosine similarity between a test embedding and the in-distribution embeddings stored in the memory bank $\mathcal{M}$, tends to be smaller for OOD samples than for in-distribution samples. Consequently, the perturbation factor $1 - \lambda$ becomes smaller for in-distribution samples, inducing a lower entropy increase relative to OOD and thereby resolving cases in which conventional likelihood-based detection fails. Conversely, when the in-distribution has lower entropy than the OOD, the manipulation by $\lambda$ effectively enlarges the entropy discrepancy between the two distributions, thereby improving likelihood separability and leading to stronger detection performance. Furthermore, we formally derive how entropy manipulation through SPEM modifies the expected likelihood difference between in-distribution and OOD samples under a density estimation model, and establish a lower bound on this difference as presented in Theorem 4.1. The proof of this theorem is reported in Appendix A.

**Theorem 4.1.** *Let $P$, $P_\theta$, and $Q$ be $d$-dimensional continuous probability distributions on $\mathbb{R}^d$. Let*

$$X \sim P, \quad Y \sim Q, \quad Z \sim \mathcal{N}(0, \sigma_P^2 I_d), \quad Z' \sim \mathcal{N}(0, \sigma_Q^2 I_d),$$

*and define*

$$X + Z \sim P', \quad Y + Z' \sim Q'.$$

*Let $P$ have covariance matrix $\Sigma$. Further assume that $\sigma_P$ and $\sigma_Q$ are positive random variables, each drawn independently from a continuous probability distribution supported on $(0, \infty)$. Then, the lower bound on the expected log-likelihood difference estimated by $P_\theta$ between $P'$ and $Q'$ is*

$$\frac{d}{2}\left(\log\left(\frac{e^{\frac{2}{d}\mathbb{H}(Y)} + 2\pi e(e^{\mathbb{E}[\log \sigma_Q^2]})}{2\pi e(\Pi_{i=1}^d(\lambda_i + \mathbb{E}[\sigma_P^2]))^{\frac{1}{d}}}\right)\right) - D_{KL}(P'\|P_\theta)$$

*where $\lambda_i$ denotes the $i$-th eigenvalue of $\Sigma$.*

Theorem 4.1 demonstrates that, if the perturbation applied to $Q$ is sufficiently stronger than that applied to $P$ (i.e., generally $\sigma_Q > \sigma_P$), then the lower bound of the expected log-likelihood difference can be increased. Additionally, if the feature extractor $g$ is well trained, the similarity between $g(\mathbf{x}_{\text{test}})$ and the in-distribution embeddings stored in the memory bank naturally yields a smaller value of $\sigma_P$ compared to $\sigma_Q$ for OOD-like inputs. Theorem 4.1 provides theoretical evidence that SPEM enlarges the expected log-likelihood difference between in-distribution and OOD.

## 5 EXPERIMENT

In this section, we compare the OOD detection performance of SPEM on real world datasets against other likelihood-based approaches.

**Data Preprocessing** We utilized CIFAR-10, CIFAR-100, SVHN, CelebA, MNIST (Deng, 2012), and FashionMNIST (Xiao et al., 2017) datasets provided by torchvision (Marcel & Rodriguez, 2010), which are widely adopted benchmark datasets in likelihood-based OOD detection methods. From these, we constructed five dataset pairs, each consisting of one in-distribution dataset and one OOD dataset, and we also evaluated the reversed pairing. In total, we evaluated OOD detection performance across ten in/out-of-distribution dataset pairs. All images were resized to $32 \times 32$, and the train/test splits provided by torchvision were adopted for each dataset. We applied uniform dequantization by adding pixel-wise noise $u \sim U(0, 1/256)$ to the inputs, enabling them to be interpreted as continuous distributions. Since we employed models pretrained on ImageNet, which generally take three-channel images as input, we converted MNIST and FashionMNIST to three-channel format by replicating the single channel across all three-channels. In our experiments, the detection performance was measured using AUROC.

**Likelihood-based Models** All comparison models except the local intrinsic dimension-based method are based on ResFlow, a normalizing flow model commonly used in the image domain, with latent distribution $\mathcal{N}(0, I_d)$, and we compare our proposed method with the following seven approaches:

1. Likelihood only. For a test vector $\mathbf{x}$, we directly use negative log-likelihood $-\log p(\mathbf{x})$ as the anomaly score for OOD detection.

2. Complexity-based method (Serrà et al., 2019). We measure the bit length $L(\mathbf{x})$ by compressing the image using PNG and obtaining the resulting number of bits. The OOD score is then defined as $S(\mathbf{x}) = -\log p(\mathbf{x}) - L(\mathbf{x})$, which incorporates the complexity term into the likelihood.

3. Typicality-based method ($p(\mathbf{z})$) (Nalisnick et al., 2019). Since the latent of the normalizing flow is assumed to follow $N(0, I_d)$, when the in-distribution is $P$, we use $S(\mathbf{z}') = |\mathbb{E}_P[\log p_{\mathcal{N}(0, I_d)}(\mathbf{z})] - \log p_{\mathcal{N}(0, I_d)}(\mathbf{z}')|$ as the OOD score, which measures how far the latent is from the latent's typicality set where $\mathbf{z}$ is a vector corresponding to $\mathbf{x}$ through flow.

4. Typicality-based method (Entropy) (Nalisnick et al., 2019). Similar to typicality test using latent, we define $S(\mathbf{x}') = |\mathbb{E}_P[\log p(\mathbf{x})] - \log p(\mathbf{x}')|$ as OOD score, which measures how far the input data $\mathbf{x}'$ falls from the input data's typicality set.

5. Likelihood ratio method (Ren et al., 2019). We define the OOD score as the likelihood ratio score $S(\mathbf{x}) = -(\log p_\theta(\mathbf{x}) - \log p_{\theta_b}(\mathbf{x}))$, where $p_\theta$ denotes the model trained on the in-distribution and $p_{\theta_b}$ is the background-trained model constructed by introducing perturbations at the pixel level.

6. GMM method using various statistics (Osada et al., 2024). We train a Gaussian Mixture Model (GMM) using the latent log-likelihood $\log p(\mathbf{z})$ and the bit-length $L(\mathbf{x})$ of the input data $\mathbf{x}$ obtained from a general compressor such as PNG, and employ the negative log-likelihood estimated by the GMM as the OOD score.

7. Local intrinsic dimension-based method Kamkari et al. (2024). This method performs OOD detection using a dual threshold based on local intrinsic dimension (LID) and log-likelihood. While the original formulation requires computing the Jacobian of the flow, the computational cost makes a faithful re-implementation impractical. To ensure comparability, we therefore report the AUROC values provided in the original paper. Since the official implementation of LID relies on a validation set and a slightly different data-split protocol, we treat the reported AUROC as a reference upper bound rather than a strictly comparable baseline. Nevertheless, including LID in our tables enables a fairer contextualization against established state-of-the-art methods.

SPEM's feature extractor used a ResNet-152 (He et al., 2016), pretrained on ImageNet, and the hyperparameter $\alpha$, which controls the manipulation intensity, was set to 0.4 in all experimental settings. Additionally, to examine the influence of the semantic information contained in the original

Table 1: OOD detection performance on real image dataset using ResFlow. The five dataset pairs on the left are widely recognized as cases where likelihood-based OOD detection fails, and the remaining pairs do not exhibit problematic behavior in the estimated likelihood. For each dataset pair, the values that ranked within the top two across all compared models are highlighted in bold.

| $P$ (In) | CIFAR-10 | CIFAR-100 | CelebA | CIFAR-10 | FashionMNIST | SVHN | SVHN | SVHN | CelebA | MNIST |
| $Q$ (Out) | SVHN | SVHN | SVHN | CelebA | MNIST | CIFAR-10 | CIFAR-100 | CelebA | CIFAR-10 | FashionMNIST |
|---|---|---|---|---|---|---|---|---|---|---|
| Likelihood | 0.0256 | 0.0277 | 0.0163 | 0.6346 | 0.8797 | 0.9967 | 0.9950 | 0.9988 | 0.5819 | 0.9786 |
| Complexity | 0.7943 | 0.7331 | 0.7998 | 0.6252 | 0.8811 | 0.9912 | 0.9890 | 0.9983 | 0.5740 | 0.3411 |
| Typicality ($p(\mathbf{z})$) | 0.0226 | 0.0273 | 0.0306 | 0.5429 | 0.7914 | 0.9969 | 0.9954 | 0.9987 | 0.6628 | 0.9954 |
| Typicality (Entropy) | 0.8866 | 0.8783 | 0.9489 | 0.3633 | 0.7703 | 0.9938 | 0.9893 | 0.9987 | 0.6082 | 0.9688 |
| Likelihood Ratio | 0.8512 | 0.7952 | 0.4458 | 0.7018 | **0.9718** | 0.9640 | 0.9418 | 0.9962 | 0.1826 | 0.0874 |
| GMM | 0.8522 | 0.8119 | 0.9128 | 0.4089 | 0.9267 | 0.9960 | 0.9958 | 0.9975 | 0.6806 | 0.9953 |
| LID | 0.9360 | 0.9330 | 0.9490 | 0.6550 | **0.9510** | 0.9870 | 0.9860 | 0.9960 | 0.9390 | **1.0000** |
| SPEM | **0.9913** | **0.9359** | **1.0000** | **0.9830** | 0.9207 | **0.9997** | **0.9992** | **1.0000** | **0.9999** | **1.0000** |
| SPEM-noise | **0.9943** | **0.9458** | **1.0000** | **0.9873** | 0.9432 | **0.9997** | **0.9990** | **1.0000** | **0.9999** | **1.0000** |

images, we introduced SPEM-noise as a comparative model, which uses only the noise derived through similarity as input to the density model. In this setting, the hyperparameter $\alpha$ was fixed at 0.1. Similarly to the other comparison models, ResFlow was used as the density estimation model. The implementation details of the comparison models and SPEM are described in Appendix B.

**Experiment Result** According to Table 1, SPEM consistently outperforms other comparison methods across most in/out-of-distribution pairs. Moreover, while other methods (except LID) fail to provide consistently stable detection performance, SPEM achieves robust performance regardless of the entropy ordering between in-distribution and out-of-distribution data. Interestingly, we find that SPEM-noise, which does not incorporate the original test data, outperforms SPEM. This result implies that superior performance can be achieved on Gaussian images generated solely in proportion to similarity values that encode semantic information, while disregarding the entropies of the original distributions, $\mathbb{H}(P)$ and $\mathbb{H}(Q)$. This suggests that the semantic information inherent in the original images does not determine the likelihood ordering; rather, the entropy of the distribution to which the images belong may introduce confounding effects into the likelihood ordering. To further substantiate this observation, we include in Appendix C an ablation study that analyses not only the performance differences arising from the expressive power of the feature extractor used in SPEM and the effectiveness of ReAct in refining the embedding vector, but also the reasons why SPEM-noise can outperform SPEM in certain settings.

Table 2: OOD detection performance on real image dataset using Glow. The five dataset pairs on the left are widely recognized as cases where likelihood-based OOD detection fails, and the remaining pairs do not exhibit problematic behavior in the estimated likelihood. For each dataset pair, the values that ranked within the top two across all compared models are highlighted in bold.

| $P$ (In) | CIFAR-10 | CIFAR-100 | CelebA | CIFAR-10 | FashionMNIST | SVHN | SVHN | SVHN | CelebA | MNIST |
| $Q$ (Out) | SVHN | SVHN | SVHN | CelebA | MNIST | CIFAR-10 | CIFAR-100 | CelebA | CIFAR-10 | FashionMNIST |
|---|---|---|---|---|---|---|---|---|---|---|
| Likelihood | 0.0807 | 0.0939 | 0.1292 | 0.5144 | 0.7359 | 0.9919 | 0.9906 | 0.9991 | 0.7408 | 0.9997 |
| Complexity | 0.8718 | 0.8319 | 0.9737 | 0.5566 | 0.8604 | 0.5270 | 0.6020 | 0.6163 | 0.7564 | 0.9712 |
| Typicality ($p(\mathbf{z})$) | 0.6530 | 0.6764 | 0.9998 | 0.6297 | 0.7365 | 0.9783 | 0.9744 | 0.9976 | 0.9253 | 0.9997 |
| Typicality (Entropy) | 0.4929 | 0.4762 | 0.7300 | 0.4283 | 0.5878 | 0.9894 | 0.9876 | 0.9990 | 0.7237 | 0.9997 |
| Likelihood Ratio | 0.8477 | 0.6020 | 0.8550 | 0.7334 | **0.9616** | 0.1078 | 0.1398 | 0.1548 | 0.5146 | 0.9875 |
| GMM | 0.8744 | 0.8421 | **0.9997** | 0.4600 | 0.8896 | 0.9762 | 0.9747 | 0.9943 | 0.8646 | 0.9996 |
| LID | 0.9360 | **0.9330** | 0.9490 | 0.6550 | **0.9510** | 0.9870 | 0.9860 | 0.9960 | 0.9390 | **1.0000** |
| SPEM | **0.9768** | 0.8655 | 0.9697 | **0.9782** | 0.9424 | **0.9985** | **0.9975** | **0.9993** | **0.9701** | **1.0000** |
| SPEM-noise | **0.9916** | **0.9343** | **0.9999** | **0.9830** | 0.9424 | **0.9996** | **0.9988** | **0.9999** | **0.9993** | **1.0000** |

**Model-Agnostic Evaluation** We repeat the protocol with Glow in place of ResFlow. As shown in Table 2, SPEM performs better than other likelihood-based detectors on most pairs, including the difficult case where the in-distribution has higher entropy than the OOD. This indicates that the effect is not tied to a specific flow architecture (see Appendix C for details).

## 6 DISCUSSION

**Meaning of Likelihood** Building upon the empirical success of SPEM and the entropy-based theoretical framework, a natural question arises: *what does model likelihood measure in practice?* As shown in Section 5, our results with SPEM indicates that regions assigned high likelihood are often those where a lower-entropy distribution dominates the comparison, rather than regions defined by semantic similarity. Thus, the likelihood value itself does not encode semantics; instead, for a given

input it tends to reflect how concentrated the underlying distribution is. This observation is consistent with the phenomenon of likelihood inversion, whereby an OOD—despite being semantically distinct from the in-distribution—can attain higher likelihood values when its entropy is sufficiently lower.

**Sensitivity of Likelihood** To examine the performance trend with respect to changes in the entropy of the OOD , we conducted the experiment illustrated in Figure 3. After training Glow on a real image dataset, we set the OOD to a Gaussian $\mathcal{N}(0, \sigma^2 I_d)$ and evaluated AUROC using only log-likelihood for detection as $\sigma$ varied. Unlike the SPEM framework, which injects noise into OOD samples composed of real images, this experiment directly manipulates the entropy of an OOD consisting purely of noise. As shown in Figure 3, the AUROC for SVHN rises steeply at small values of $\sigma$, reaching high performance earlier than CIFAR-10 and CIFAR-100. Interestingly, CelebA follows a pattern much closer to SVHN than to CIFAR-10: its AUROC also escalates at similar noise levels, despite CelebA having entropy comparable to CIFAR-10. This discrepancy indicates that the observed sensitivity cannot be explained by entropy differences alone. While the convergence of AUROC toward 1 as $\sigma$ increases across all datasets suggests that relative

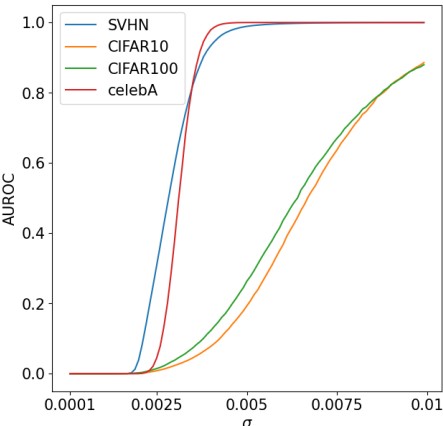

Figure 3: AUROC for in-distributions composed of real images as a function of $\sigma$ when the OOD is $\mathcal{N}(0, \sigma^2 I_d)$.

entropy differences strongly affect likelihood ordering, this phenomenon can be further understood through the role of the KL-divergence $D_{KL}(Q||P_\theta)$ in Equation 1. In future work, we expect that if a likelihood-based detection methodology that accounts for not only the entropic aspect of the in-distribution but also the KL-divergence term $D_{\mathrm{KL}}(Q||P_\theta)$, it will yield more reliable detection performance with density estimation models. Since computing $D_{\mathrm{KL}}(Q||P_\theta)$ is generally infeasible in practice, such approaches would need to exploit prior knowledge about the specific cases where likelihood-based methods fail as inductive biases or auxiliary signals to guide the generative model's design.

**Hyperparameter Robustness** Figure 4 shows that AUROC increases as $\alpha$ grows and then plateaus; around $\alpha \approx 0.4$ performance is close to its maximum and remains stable to small changes. Beyond this region, additional gains are limited and very large $\alpha$ can introduce numerical issues in the flow, so a "sufficiently large but not excessive" value is recommended (we used $\alpha$ chosen in this stable band). These trends hold across the tested ID/OOD pairs and indicate that the method is not overly sensitive to precise tuning of $\alpha$ (see Appendix C.1 for details).

**Insight from SPEM-Noise** SPEM-Noise, which discards the original image and feeds only similarity-scaled Gaussian noise, can match or even surpass SPEM. This points to the flow likelihood being more sensitive to distributional concentration (e.g., effective spread/entropy) than to high-level semantics, a pattern consistent with our broader observations; we provide supporting results and analysis in Appendix C.5.

# 7 CONCLUSION

We proposed SPEM to mitigate the likelihood paradox arising from likelihood assignment in generative models. From an entropic perspective, we explained why SPEM succeeds not only in cases where conventional likelihood-based detection methods fail in OOD detection but also in scenarios where they already perform well. Furthermore, we empirically demonstrated on real-world datasets that our method outperforms existing baselines that rely solely on generative model likelihoods. Since our method as well as other approaches in this research field still requires auxiliary processes, we anticipate future work to develop training methodologies that align the likelihood assignment of generative models with human intuition, enabling likelihood-based OOD detection without auxiliary procedures. Through our work, we hope this study will stimulate further investigation into how to interpret and utilize likelihood estimates produced by generative models.

## REPRODUCIBILITY STATEMENT

To ensure the reproducibility of our findings and to promote transparency in our research, we provide a detailed account of the experimental setup used throughout the study. In particular, Appendix B presents comprehensive information that is essential for faithfully replicating our results, including explicit specifications of the hyperparameters employed, the training configurations, and other implementation details that could potentially affect model performance. Beyond the written description, we also supply the source code in the supplementary materials. This not only enables exact reproduction of the experiments reported in the paper but also serves as a resource for practitioners and researchers who wish to extend, modify, or apply our methods to related problems in future work.

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

## A    PROOF OF THEOREMS

**Theorem A.1.** *Let $P$, $P_\theta$, $Q$, be d-dimensional continuous probability distributions on $\mathbb{R}^d$. Let $X \sim Q$, $Z \sim \mathcal{N}(0, \sigma^2 I_d)$, and define $Q'$ as the distribution of $X + Z$. Then a lower bound on the expected log-likelihood difference estimated by $P_\theta$ between $P$ and $Q'$ is*

$$\frac{d}{2} \log \left( e^{\frac{2}{d} \mathbb{H}(X)} + 2\pi e \sigma^2 \right) - \mathbb{H}(P) - D_{KL}(P || P_\theta).$$

*Proof.* The difference between the log-likelihood expectations estimated by $P_\theta$ for $P$ and $Q$ can be derived as follows:

$$
\begin{aligned}
& \mathbb{E}_{\mathbf{x} \sim P}[\log P_\theta(\mathbf{x})] - \mathbb{E}_{\mathbf{x} \sim Q}[\log P_\theta(\mathbf{x})] \\
&= -D_{KL}(P || P_\theta) - \mathbb{H}(P) + D_{KL}(Q || P_\theta) + \mathbb{H}(Q) \\
&= D_{KL}(Q || P_\theta) - D_{KL}(P || P_\theta) + \mathbb{H}(Q) - \mathbb{H}(P)
\end{aligned}
\tag{4}
$$

Therefore, if we rewrite this equation by modifying $Q$ to $Q'$,

$$
\begin{aligned}
& \mathbb{E}_{\mathbf{x} \sim P}[\log P_\theta(\mathbf{x})] - \mathbb{E}_{\mathbf{x} \sim Q'}[\log P_\theta(\mathbf{x})] \\
&= D_{KL}(Q' || P_\theta) - D_{KL}(P || P_\theta) + \mathbb{H}(Q') - \mathbb{H}(P) \\
&\geq -D_{KL}(P || P_\theta) + \mathbb{H}(Q') - \mathbb{H}(P)
\end{aligned}
\tag{5}
$$

If we express $\mathbb{H}(Q')$ in terms of a random variable rather than a distribution, we get the following:

$$\mathbb{H}(Q') = \mathbb{H}(X + Z) \tag{6}$$

The entropy power of $X$ is defined as (Cover, 1999):

$$N(X) = \frac{1}{2\pi e} e^{\frac{2}{d} \mathbb{H}(X)} \tag{7}$$

Since $X$ and $Z$ are independent, the following formula holds due to entropy power inequality:

$$
\begin{aligned}
& \frac{1}{2\pi e} e^{\frac{2}{d} \mathbb{H}(X+Z)} \geq \frac{1}{2\pi e} \left( e^{\frac{2}{d} \mathbb{H}(X)} + e^{\frac{2}{d} \mathbb{H}(Z)} \right) \\
& \Rightarrow e^{\frac{2}{d} \mathbb{H}(X+Z)} \geq e^{\frac{2}{d} \mathbb{H}(X)} + e^{\frac{2}{d} \mathbb{H}(Z)} \\
& \Rightarrow \mathbb{H}(X + Z) \geq \frac{d}{2} \log \left( e^{\frac{2}{d} \mathbb{H}(X)} + 2\pi e \sigma^2 \right)
\end{aligned}
\tag{8}
$$

Therefore, we can derive follows:

$$
\begin{aligned}
& \mathbb{E}_{\mathbf{x} \sim P}[\log P_\theta(\mathbf{x})] - \mathbb{E}_{\mathbf{x} \sim Q'}[\log P_\theta(\mathbf{x})] \\
&\geq -D_{KL}(P || P_\theta) + \mathbb{H}(Q') - \mathbb{H}(P) \\
&\geq \frac{d}{2} \log \left( e^{\frac{2}{d} \mathbb{H}(X)} + 2\pi e \sigma^2 \right) - \mathbb{H}(P) - D_{KL}(P || P_\theta)
\end{aligned}
\tag{9}
$$

$\square$

**Theorem A.2.** *Let $P$, $P_\theta$, and $Q$ be d-dimensional continuous probability distributions on $\mathbb{R}^d$. Let*

$$X \sim P, \quad Y \sim Q, \quad Z \sim \mathcal{N}(0, \sigma_P^2 I_d), \quad Z' \sim \mathcal{N}(0, \sigma_Q^2 I_d),$$

*and define*

$$X + Z \sim P', \quad Y + Z' \sim Q'.$$

*Let $P$ have covariance matrix $\Sigma$. Further assume that $\sigma_P$ and $\sigma_Q$ are positive random variables, each drawn independently from a continuous probability distribution supported on $(0, \infty)$. Then, the lower bound on the expected log-likelihood difference estimated by $P_\theta$ between $P'$ and $Q'$ is*

$$\frac{d}{2}\left(\log\left(\frac{e^{\frac{2}{d}\mathbb{H}(Y)} + 2\pi e(e^{\mathbb{E}[\log\sigma_Q^2]})}{2\pi e(\Pi_{i=1}^d(\lambda_i + \mathbb{E}[\sigma_P^2]))^{\frac{1}{d}}}\right)\right) - D_{KL}(P'||P_\theta)$$

where $\lambda_i$ denotes the $i$-th eigenvalue of $\Sigma$.

*Proof.* Since $Y$ and $Z'$ are independent, the following formula holds due to entropy power inequality:

$$\mathbb{H}(Q') = \mathbb{H}(Y + Z') \geq \frac{d}{2}\log(e^{\frac{2}{d}\mathbb{H}(Y)} + e^{\frac{2}{d}\mathbb{H}(Z')}) \tag{10}$$

We can expand $\mathbb{H}(Z')$ as follows using the property of conditional entropy:

$$\begin{aligned}
\mathbb{H}(Z') &= \mathbb{E}\left[\mathbb{H}(Z'|\sigma_Q)\right] + \mathcal{I}(Z';\sigma_Q) \\
&= \frac{d}{2}(\log(2\pi e) + \mathbb{E}[\log\sigma_Q^2]) + \mathcal{I}(Z';\sigma_Q) \ (\because \mathbb{H}(Z'|\sigma_Q) = \frac{d}{2}\log(2\pi e\sigma_Q^2))
\end{aligned} \tag{11}$$

Therefore, we can rewrite lower bound of $\mathbb{H}(Y + Z')$ as follows:

$$\begin{aligned}
\mathbb{H}(Y + Z') &\geq \frac{d}{2}\log(e^{\frac{2}{d}\mathbb{H}(Y)} + e^{\frac{2}{d}\mathbb{H}(Z')}) \\
&= \frac{d}{2}\log(e^{\frac{2}{d}\mathbb{H}(Y)} + e^{\log(2\pi e) + \mathbb{E}[\log\sigma_Q^2] + \frac{2}{d}\mathcal{I}(Z';\sigma_Q)}) \\
&\geq \frac{d}{2}\log(e^{\frac{2}{d}\mathbb{H}(Y)} + e^{\log(2\pi e) + \mathbb{E}[\log\sigma_Q^2]}) \ (\because \mathcal{I}(Z';\sigma_Q) \geq 0) \\
&= \frac{d}{2}\log(e^{\frac{2}{d}\mathbb{H}(Y)} + 2\pi e(e^{\mathbb{E}[\log\sigma_Q^2]}))
\end{aligned} \tag{12}$$

Hence, we can derive:

$$\begin{aligned}
&\mathbb{E}_{\mathbf{x}\sim P'}[\log P_\theta(\mathbf{x})] - \mathbb{E}_{\mathbf{x}\sim Q'}[\log P_\theta(\mathbf{x})] \\
&\geq -D_{KL}(P'||P_\theta) + \mathbb{H}(Q') - \mathbb{H}(P') \\
&= \mathbb{H}(Y + Z') - \mathbb{H}(P') - D_{KL}(P'||P_\theta) \\
&= \frac{d}{2}\log(e^{\frac{2}{d}\mathbb{H}(Y)} + 2\pi e(e^{\mathbb{E}[\log\sigma_Q^2]})) - \mathbb{H}(P') - D_{KL}(P'||P_\theta)
\end{aligned} \tag{13}$$

Since $X$ and $Z$ are independent, the following holds due to the maximum entropy property of Gaussian distribution:

$$\begin{aligned}
\mathbb{H}(P') &\leq \mathbb{H}(\mathcal{N}(0, \text{Cov}(X + Z))) \\
&= \mathbb{H}(\mathcal{N}(0, \text{Cov}(X) + \text{Cov}(Z)))
\end{aligned} \tag{14}$$

Because of law of total covariance, we can derive:

$$\begin{aligned}
\mathbb{H}(P') &\leq \mathbb{H}(\mathcal{N}(0, \text{Cov}(X + Z))) \\
&= \mathbb{H}(\mathcal{N}(0, \text{Cov}(X) + \text{Cov}(Z))) \\
&= \mathbb{H}(\mathcal{N}(0, \Sigma + \mathbb{E}[\text{Cov}(Z|\sigma_P)] + \text{Cov}(\mathbb{E}[Z|\sigma_P]))) \ (\because \text{Law of Total Covariance}) \\
&= \mathbb{H}(\mathcal{N}(0, \Sigma + \mathbb{E}[\sigma_P^2]I_d)) \ (\because \mathbb{E}[Z|\sigma_P] = 0, \text{Cov}(Z|\sigma_P) = \sigma_P^2 I_d) \\
&= \frac{1}{2}\log((2\pi e)^d \det(\Sigma + \mathbb{E}[\sigma_P^2]I_d))
\end{aligned} \tag{15}$$

Therefore, we can derive follows:

$$
\begin{aligned}
&\mathbb{E}_{\mathbf{x} \sim P'}[\log P_\theta(\mathbf{x})] - \mathbb{E}_{\mathbf{x} \sim Q'}[\log P_\theta(\mathbf{x})] \\
&\geq \frac{d}{2}\left(\log(e^{\frac{2}{d}\mathbb{H}(Y)} + 2\pi e(e^{\mathbb{E}[\log \sigma_Q^2]}))\right) - \mathbb{H}(P') - D_{KL}(P'||P_\theta) \\
&\geq \frac{d}{2}\left(\log(e^{\frac{2}{d}\mathbb{H}(Y)} + 2\pi e(e^{\mathbb{E}[\log \sigma_Q^2]}))\right) - \frac{1}{2}\log((2\pi e)^d \det(\Sigma + \mathbb{E}[\sigma_P^2]I_d)) - D_{KL}(P'||P_\theta) \\
&= \frac{d}{2}\left(\log(e^{\frac{2}{d}\mathbb{H}(Y)} + 2\pi e(e^{\mathbb{E}[\log \sigma_Q^2]}))\right) - \frac{d}{2}\log(2\pi e(\det(\Sigma + \mathbb{E}[\sigma_P^2]I_d))^{\frac{1}{d}}) - D_{KL}(P'||P_\theta) \\
&= \frac{d}{2}\left(\log(e^{\frac{2}{d}\mathbb{H}(Y)} + 2\pi e(e^{\mathbb{E}[\log \sigma_Q^2]})) - \log(2\pi e(\det(\Sigma + \mathbb{E}[\sigma_P^2]I_d))^{\frac{1}{d}})\right) - D_{KL}(P'||P_\theta) \\
&= \frac{d}{2}\left(\log\left(\frac{e^{\frac{2}{d}\mathbb{H}(Y)} + 2\pi e(e^{\mathbb{E}[\log \sigma_Q^2]})}{2\pi e(\det(\Sigma + \mathbb{E}[\sigma_P^2]I_d))^{\frac{1}{d}}}\right)\right) - D_{KL}(P'||P_\theta) \\
&= \frac{d}{2}\left(\log\left(\frac{e^{\frac{2}{d}\mathbb{H}(Y)} + 2\pi e(e^{\mathbb{E}[\log \sigma_Q^2]})}{2\pi e(\Pi_{i=1}^{d}(\lambda_i + \mathbb{E}[\sigma_P^2]))^{\frac{1}{d}}}\right)\right) - D_{KL}(P'||P_\theta)
\end{aligned}
\tag{16}
$$

$\square$

## B  DETAILS OF SECTION 5

This section provides detailed information about the experimental implementation in Section 5.

For the density estimation model used in all comparison methods except LID, we employed ResFlow and utilized the library implemented by Stimper et al. (2023). As the optimizer, we used Adam (Kingma & Ba, 2014) with a weight decay of 1e-4 and a batch size of 64, and the initial learning rate was set to 1e-4. For the learning rate scheduler, we adopted CosineAnnealingWarmRestarts (Loshchilov & Hutter, 2016), setting the minimum learning rate to 1e-6. The cycle of decaying and increasing the learning rate was initially configured to half of the total number of epochs. However, since this setting caused training instability, we modified it by setting the entire number of epochs as a single cycle. In the ResFlow architecture, the number of multiscale blocks was fixed at three. For CIFAR-10/100, SVHN, and CelebA, the latent dimensionality was set to 128 and each multiscale block comprised 12 layers, while for MNIST and FashionMNIST the latent dimensionality and the number of layers per block were set to 64 and 5, respectively. In addition, the number of training epochs was set to 100 for CIFAR-10/100. The rectification threshold of ReAct $\beta$ was determined by randomly sampling 1,000 in-distribution training embedding vectors and setting the cutoff to their 90th percentile value.

When calculating the likelihood of input data, we calculated the log-likelihood using the same density model across all implemented methods. For the complexity-based method, we employed the PNG compressor available in the Python OpenCV library (Bradski, 2000) to compute the number of bits, which was then used to perform the complexity calculation. In the likelihood ratio method, the background model was trained with the same hyperparameters as those used for estimating the density of the original input data, while the hyperparameter $\alpha$, which controls the probability of pixel perturbation, was set to 0.2. For the method utilizing Gaussian Mixture Models (GMMs), we implemented the model using scikit-learn (Pedregosa et al., 2011), with the number of mixture components fixed at 3, consistent with the original paper. Pytorch 2.4.1 (Paszke et al., 2019) was used to implement the density estimation model, and the computing environment used in the experiment was AMD Ryzen 9 7950X for CPU and RTX 4090Ti for GPU.

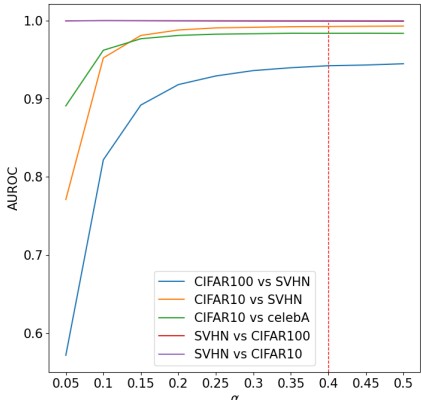

Figure 4: OOD detection performance according to $\alpha$, which controls the intensity of entropy manipulation. The legend indicates the experimental dataset pairs, with the former indicating in-distribution and the latter indicating OOD. All experiments were set up identically to those in Table 1.

## C  ABLATION STUDY OF SPEM

### C.1  HYPERPARAMETER SENSITIVITY

In SPEM, we conducted an experiment to examine how adjusting the global entropy manipulation intensity $\alpha$ affects OOD detection performance, as reported in Figure 4. According to Figure 4, when $\alpha$ is small, the entropy gap between in/out-of-distribution samples is not sufficiently enlarged, resulting in relatively low AUROC values. However, as $\alpha$ increases, the AUROC steadily improves and eventually reaches a plateau. This experiment demonstrates that by setting $\alpha$ to a sufficiently large value (e.g., $\alpha = 0.4$), the performance becomes robust to small variations around this range while achieving high AUROC. Furthermore, we observed that excessively large values of $\alpha$ intro-duce numerical stability issues in density estimation. Therefore, we recommend setting $\alpha$ to an ade-quately large but not excessive value, since additional performance gains beyond a certain threshold are limited.

### C.2  EFFECT OF FEATURE EXTRACTOR

Table 3: OOD detection performance using ResFlow according to changes in feature extractor.

| $P$ (In) | CIFAR-10 | CIFAR-100 | CelebA | CIFAR-10 | FashionMNIST | SVHN | SVHN | SVHN | CelebA | MNIST |
| $Q$ (Out) | SVHN | SVHN | SVHN | CelebA | MNIST | CIFAR-10 | CIFAR-100 | CelebA | CIFAR-10 | FashionMNIST |
|---|---|---|---|---|---|---|---|---|---|---|
| ResNet-50 | 0.9839 | 0.9435 | 0.9999 | 0.9547 | **0.9643** | 0.9991 | 0.9984 | 0.9995 | 0.9997 | 1.0000 |
| ResNet-101 | 0.9894 | **0.9536** | **1.0000** | 0.9691 | 0.8623 | 0.9996 | **0.9992** | 0.9999 | 0.9999 | **0.9999** |
| ResNet-152 | **0.9913** | 0.9359 | **1.0000** | **0.9830** | 0.9207 | **0.9997** | **0.9992** | **1.0000** | 0.9999 | **1.0000** |

Table 4: OOD detection performance using Glow according to changes in feature extractor.

| $P$ (In) | CIFAR-10 | CIFAR-100 | CelebA | CIFAR-10 | FashionMNIST | SVHN | SVHN | SVHN | CelebA | MNIST |
| $Q$ (Out) | SVHN | SVHN | SVHN | CelebA | MNIST | CIFAR-10 | CIFAR-100 | CelebA | CIFAR-10 | FashionMNIST |
|---|---|---|---|---|---|---|---|---|---|---|
| ResNet-50 | 0.9583 | 0.8712 | 0.9634 | 0.9498 | **0.9774** | 0.9965 | 0.9959 | 0.9975 | 0.9628 | **1.0000** |
| ResNet-101 | 0.9722 | **0.8897** | **0.9705** | 0.9621 | 0.8887 | 0.9979 | **0.9977** | 0.9992 | 0.9676 | 0.9999 |
| ResNet-152 | **0.9768** | 0.8655 | 0.9697 | **0.9782** | 0.9424 | **0.9985** | 0.9975 | **0.9993** | **0.9701** | **1.0000** |

We conducted experiments on the trend of OOD detection performance by changing the feature extractor that determines the similarity value $\lambda$ with embedding vector in memory bank, which controls the strength of entropy manipulation. As feature extractors, we used ResNet-50/101/152 pretrained on ImageNet, and the results are reported in Table 3 and 4. According to Table 3 and 4, ResNet-152, which has greater expressive power in our ablation study, generally achieved higher performance and thus we adopted it as the feature extractor in our main experiments. This can be interpreted as follows: the stronger the expressive power, the more pronounced the difference in similarity between in/out-of-distribution, leading to the results shown in experiments. Based on this

finding, one way to further improve the proposed SPEM is to fine-tune a model pretrained on the in-distribution dataset and then plug the tuned model into SPEM. Although this approach requires additional computational resources, it could yield even better OOD detection performance.

## C.3 EFFECT OF REACT

Table 5: Performance difference in OOD detection using ResFlow with and without the ReAct module.

| $P$ (In) | CIFAR-10 | CIFAR-100 | CelebA | CIFAR-10 | FashionMNIST | SVHN | SVHN | SVHN | CelebA | MNIST |
| $Q$ (Out) | SVHN | SVHN | SVHN | CelebA | MNIST | CIFAR-10 | CIFAR-100 | CelebA | CIFAR-10 | FashionMNIST |
| w/ ReAct | **0.9913** | **0.9359** | **1.0000** | 0.9830 | **0.9207** | **0.9997** | **0.9992** | **1.0000** | 0.9999 | **1.0000** |
| w/o ReAct | 0.9536 | 0.8471 | 0.9999 | **0.9839** | 0.9042 | 0.9994 | 0.9988 | 0.9999 | **0.9999** | **1.0000** |

Table 6: Performance difference in OOD detection using Glow with and without the ReAct module.

| $P$ (In) | CIFAR-10 | CIFAR-100 | CelebA | CIFAR-10 | FashionMNIST | SVHN | SVHN | SVHN | CelebA | MNIST |
| $Q$ (Out) | SVHN | SVHN | SVHN | CelebA | MNIST | CIFAR-10 | CIFAR-100 | CelebA | CIFAR-10 | FashionMNIST |
| w/ ReAct | **0.9768** | **0.8655** | 0.9697 | 0.9782 | **0.9424** | 0.9985 | 0.9975 | 0.9993 | 0.9701 | **1.0000** |
| w/o ReAct | 0.9116 | 0.7570 | **0.9806** | **0.9795** | 0.9284 | **0.9990** | **0.9983** | **0.9998** | **0.9886** | **1.0000** |

We incorporated ReAct, a method that rectifies the activations of the penultimate layer in pretrained models, into our proposed framework SPEM to enhance OOD detection performance. The motivation behind this integration is that we expected that the similarity gap between test vectors sampled from in/out-of-distribution and the in-distribution training vectors stored in the memory bank would be amplified, thereby improving detection performance. To validate this hypothesis, we measured the performance of SPEM with and without the ReAct module, and the results are reported in Table 5 and 6. Table 5 demonstrates that incorporating ReAct into SPEM with ResFlow consistently improves performance across the majority of dataset pairs. In contrast, Table 6 shows that for dataset pairs where likelihood-based methods already perform well, the use of ReAct within SPEM results in only a marginal decrease in performance. Nevertheless, in scenarios where likelihood-based approaches fail, ReAct provides substantial improvements, with instances of performance degradation remaining relatively minor. This observation suggests that ReAct is particularly beneficial in hard dataset pairs (i.e., when the entropy of the in-distribution exceeds that of the OOD distribution). Consistent with the findings reported in the original ReAct paper, these results provide evidence that rectification amplifies the separation between in-distribution and OOD embedding vectors, thereby highlighting the difference in similarity and ultimately improving detection performance.

## C.4 PERFORMANCE COMPARISON OF SPEM USING GLOW

To examine the compatibility and performance consistency of SPEM with different density estimation models, we additionally employed Glow to estimate the in-distribution and compared its performance. Most experimental settings were aligned with those of ResFlow, with the exception that for CIFAR-10/100, SVHN, and CelebA, the latent dimensionality was set to 256 and each multiscale block comprised 16 layers, while for MNIST and FashionMNIST the latent dimensionality and the number of layers per block were set to 64 and 5, respectively. The number of training epochs was fixed at 300 for CIFAR-10/100, SVHN, and CelebA, and 80 for MNIST and FashionMNIST. For SPEM, $\alpha$ was set to 0.3, and for SPEM-noise it was set to 0.1. The experimental results are reported in Table 2.

As shown in Table 2, SPEM with Glow consistently outperforms other comparison methods across most in/out-of-distribution pairs. Methods that rely solely on likelihood yield extremely low AU-ROC when the in-distribution has higher entropy than the OOD distribution (e.g., when SVHN is used as the in-distribution). In these cases, both the comparison methods and our proposed approach clearly surpass the likelihood-only baseline. Conversely, when the in-distribution entropy is lower than that of the OOD, some comparison methods perform worse than the likelihood baseline. Nevertheless, our approach consistently outperforms all comparison methods, even in these challenging scenarios. In addition, we observed that the performance gap between SPEM and SPEM-noise was larger when using Glow compared to ResFlow. This result suggests that the probability models estimated by different density estimators are inherently distinct. Although SPEM using Glow achieved

slightly lower performance than when using ResFlow, the fact that it still outperformed most comparison models demonstrates that SPEM's superiority in likelihood-based OOD detection is largely agnostic to the choice of the underlying density model.

## C.5  ANALYSIS OF SPEM-NOISE

In the OOD detection results reported in Tables 1 and 2 SPEM-noise performs comparably to and in several cases outperforms SPEM. SPEM-noise disregards the original test data and instead measures the log-likelihood by feeding into the density estimation model only Gaussian noise, the magnitude of which is determined by the maximum similarity between the test vector and the in-distribution embedding vectors stored in the memory bank. Although this input contains no substantive semantic information, SPEM-noise nevertheless achieves performance comparable to or even superior to SPEM, which may appear counterintuitive. We account for this phenomenon by analyzing the increment in the expected log-likelihoods of in/out-of-distribution samples under SPEM-noise and SPEM. Let the in-distribution be denoted by $X \sim P$, OOD by $Y \sim Q$ that has covariance matrix $\Sigma_Q$, the noise distribution applied to the in-distribution under SPEM by $Z \sim \mathcal{N}(0, \sigma_P^2 I_d)$, and the noise distribution applied to the OOD by $Z' \sim \mathcal{N}(0, \sigma_Q^2 I_d)$. Also, we denote the density estimation model by $P_\theta$ and the manipulated in/out-of-distribution induced by the noise be denoted as $X + Z \sim P'$ and $Y + Z' \sim Q'$, respectively. In the implementation of SPEM, the noise is constructed to depend on similarity, so that $\sigma_P^2$ and $\sigma_Q^2$ in practice follow specific distributions. However, for analytical tractability, we set them as constants in our analysis in this section. Then, the difference in the expected log-likelihoods computed under SPEM can be derived as follows:

$$
\begin{aligned}
&\mathbb{E}_{\mathbf{x} \sim P'}[\log P_\theta(\mathbf{x})] - \mathbb{E}_{\mathbf{x} \sim Q'}[\log P_\theta(\mathbf{x})] \\
&= D_{KL}(Q' || P_\theta) - D_{KL}(P' || P_\theta) + \mathbb{H}(Q') - \mathbb{H}(P') \\
&= D_{KL}(Y + Z' || P_\theta) - D_{KL}(X + Z || P_\theta) + \mathbb{H}(Y + Z') - \mathbb{H}(X + Z)
\end{aligned}
$$

In addition, for the case of SPEM-noise, the difference in the expected log-likelihoods can be derived as follows:

$$
\begin{aligned}
&\mathbb{E}_{\mathbf{x} \sim Z}[\log P_\theta(\mathbf{x})] - \mathbb{E}_{\mathbf{x} \sim Z'}[\log P_\theta(\mathbf{x})] \\
&= D_{KL}(Z' || P_\theta) - D_{KL}(Z || P_\theta) + \mathbb{H}(Z') - \mathbb{H}(Z)
\end{aligned}
$$

Therefore, under SPEM-noise, the log-likelihood difference increment relative to that of SPEM denoted by $\Delta$, can be derived as follows:

$$
\begin{aligned}
\Delta &= \mathbb{E}_{\mathbf{x} \sim Z}[\log P_\theta(\mathbf{x})] - \mathbb{E}_{\mathbf{x} \sim Z'}[\log P_\theta(\mathbf{x})] - (\mathbb{E}_{\mathbf{x} \sim P'}[\log P_\theta(\mathbf{x})] - \mathbb{E}_{\mathbf{x} \sim Q'}[\log P_\theta(\mathbf{x})]) \\
&= D_{KL}(Z' || P_\theta) - D_{KL}(Z || P_\theta) + \mathbb{H}(Z') - \mathbb{H}(Z) \\
&\quad - (D_{KL}(Y + Z' || P_\theta) - D_{KL}(X + Z || P_\theta) + \mathbb{H}(Y + Z') - \mathbb{H}(X + Z)) \\
&= D_{KL}(Z' || P_\theta) - D_{KL}(Z || P_\theta) - (D_{KL}(Y + Z' || P_\theta) - D_{KL}(X + Z || P_\theta)) \\
&\quad + \mathbb{H}(Z') - \mathbb{H}(Z) - (\mathbb{H}(Y + Z') - \mathbb{H}(X + Z))
\end{aligned}
$$

We decompose this into the increment of the entropy term denoted by $\Delta_E$, and the increment of the KL-divergence denoted by $\Delta_{KL}$, and express it using the following notation:

$$
\begin{aligned}
\Delta_{KL} &= D_{KL}(Z' || P_\theta) - D_{KL}(Z || P_\theta) - (D_{KL}(Y + Z' || P_\theta) - D_{KL}(X + Z || P_\theta)) \\
\Delta_E &= \mathbb{H}(Z') - \mathbb{H}(Z) - (\mathbb{H}(Y + Z') - \mathbb{H}(X + Z))
\end{aligned}
$$

To analyze these increments, we first examine the conditions under which $\Delta_E$ becomes positive, so we derive Theorem C.1.

**Theorem C.1.** *Let $P$, $P_\theta$, and $Q$ be $d$-dimensional continuous probability distributions on $\mathbb{R}^d$. Let*

$$
X \sim P, \quad Y \sim Q, \quad Z \sim \mathcal{N}(0, \sigma_P^2 I_d), \quad Z' \sim \mathcal{N}(0, \sigma_Q^2 I_d),
$$

*and define*

$$X + Z \sim P', \quad Y + Z' \sim Q'.$$

*Let $Q$ have covariance matrix $\Sigma_Q$. Then, the sufficient condition of $\Delta_E > 0$ is*

$$\frac{\sigma_Q^2}{\sigma_P^2} > \frac{2\pi e \, (tr(\Sigma_Q))}{d e^{\frac{2}{d}\mathbb{H}(X)}}.$$

*Proof.* By applying the entropy power inequality and the maximum-entropy property of the Gaussian distribution, both of which are used in the proof of Theorem 4.1, we can derive the following results:

$$
\begin{aligned}
\mathbb{H}(Z') - \mathbb{H}(Z) &= \frac{d}{2}\log\frac{\sigma_Q^2}{\sigma_P^2} \\
\mathbb{H}(Y + Z') &\le \frac{d}{2}\log(2\pi e(\det(\Sigma_Q + \sigma_Q^2 I_d))^{\frac{1}{d}}) \\
\mathbb{H}(X + Z) &\ge \frac{d}{2}\log(e^{\frac{2}{d}\mathbb{H}(X)} + 2\pi e \sigma_P^2)
\end{aligned}
\tag{17}
$$

Therefore, we can derive follows:

$$
\begin{aligned}
\Delta_E &= \mathbb{H}(Z') - \mathbb{H}(Z) - (\mathbb{H}(Y + Z') - \mathbb{H}(X + Z)) \\
&= \frac{d}{2}\log\frac{\sigma_Q^2}{\sigma_P^2} - (\mathbb{H}(Y + Z') - \mathbb{H}(X + Z)) \\
&\ge \frac{d}{2}\log\frac{\sigma_Q^2}{\sigma_P^2} - \frac{d}{2}\log(2\pi e(\det(\Sigma_Q + \sigma_Q^2 I_d))^{\frac{1}{d}}) + \frac{d}{2}\log(e^{\frac{2}{d}\mathbb{H}(X)} + 2\pi e\sigma_P^2) \\
&= \frac{d}{2}\log\frac{\sigma_Q^2}{\sigma_P^2} - \frac{d}{2}\left(\log\frac{2\pi e(\det(\Sigma_Q + \sigma_Q^2 I_d))^{\frac{1}{d}}}{e^{\frac{2}{d}\mathbb{H}(X)} + 2\pi e\sigma_P^2}\right)
\end{aligned}
\tag{18}
$$

Finally, we derive the following sufficient condition for $\Delta_E > 0$:

$$
\begin{aligned}
\Delta_E &= \frac{d}{2}\log\frac{\sigma_Q^2}{\sigma_P^2} - \frac{d}{2}\left(\log\frac{2\pi e(\det(\Sigma_Q + \sigma_Q^2 I_d))^{\frac{1}{d}}}{e^{\frac{2}{d}\mathbb{H}(X)} + 2\pi e\sigma_P^2}\right) > 0 \\
&\Rightarrow \frac{d}{2}\log\frac{\sigma_Q^2}{\sigma_P^2} > \frac{d}{2}\left(\log\frac{2\pi e(\det(\Sigma_Q + \sigma_Q^2 I_d))^{\frac{1}{d}}}{e^{\frac{2}{d}\mathbb{H}(X)} + 2\pi e\sigma_P^2}\right) \\
&\Rightarrow \frac{\sigma_Q^2}{\sigma_P^2} > \frac{2\pi e(\det(\Sigma_Q + \sigma_Q^2 I_d))^{\frac{1}{d}}}{e^{\frac{2}{d}\mathbb{H}(X)} + 2\pi e\sigma_P^2}
\end{aligned}
\tag{19}
$$

A more conservative variance ratio inequality can be derived by eliminating both $\sigma_P^2$ and $\sigma_Q^2$ from the RHS of the inequality, first we derive upper bound of determinant using AM-GM inequality:

$$
\begin{aligned}
(\det(\Sigma_Q + \sigma_Q^2 I_d))^{\frac{1}{d}} &= (\Pi_{i=1}^d(\lambda_i + \sigma_Q^2))^{\frac{1}{d}} \\
&\le \frac{1}{d}(\Sigma_{i=1}^d(\lambda_i + \sigma_Q^2)) \\
&= \frac{\text{tr}(\Sigma_Q)}{d} + \sigma_Q^2
\end{aligned}
\tag{20}
$$

where $\lambda_i$ is $i$-th eigenvalue of $\Sigma_Q$. Then, we derive the following conservative sufficient condition for $\Delta_E > 0$:

$$\frac{\sigma_Q^2}{\sigma_P^2} > \frac{2\pi e\left(\frac{\text{tr}(\Sigma_Q)}{d} + \sigma_Q^2\right)}{e^{\frac{2}{d}\mathbb{H}(X)} + 2\pi e\sigma_P^2}$$

$$\Rightarrow \sigma_Q^2 e^{\frac{2}{d}\mathbb{H}(X)} + \sigma_P^2\sigma_Q^2 2\pi e > \sigma_P^2 2\pi e\left(\frac{\text{tr}(\Sigma_Q)}{d}\right) + \sigma_P^2\sigma_Q^2 2\pi e \tag{21}$$

$$\Rightarrow \sigma_Q^2 e^{\frac{2}{d}\mathbb{H}(X)} > \sigma_P^2 2\pi e\left(\frac{\text{tr}(\Sigma_Q)}{d}\right)$$

$$\Rightarrow \frac{\sigma_Q^2}{\sigma_P^2} > \frac{2\pi e\left(\text{tr}(\Sigma_Q)\right)}{de^{\frac{2}{d}\mathbb{H}(X)}}$$

$\square$

This implies that, regardless of the entropy gap between the in-distribution and the out-of-distribution, it is always possible to find a ratio of $\sigma_Q^2$ to $\sigma_P^2$ that ensures $\Delta_E > 0$. Additionally, Theorem C.4 shows that $\Delta_E$ is monotonically increasing in $\sigma_Q^2$, and it is monotonically decreasing in $\sigma_P^2$ with Fisher information existence assumption.

**Lemma C.2** (De Bruijn's Identity, Guo et al. (2005)). *Let $X \in \mathbb{R}^d$ be a random vector with a well-defined density, and let $Z \sim \mathcal{N}(0, I_d)$ be an independent standard Gaussian random vector. For $t > 0$, define*

$$X_t = X + \sqrt{t}Z.$$

*Then the entropy of $X_t$ satisfies*

$$\frac{d}{dt}\mathbb{H}(X_t) = \frac{1}{2}\text{tr}(\boldsymbol{J}(X_t)),$$

*where*

$$\boldsymbol{J}(X_t) = \mathbb{E}\left[\nabla \log f_{X_t}(X_t)\, \nabla \log f_{X_t}(X_t)^{\top}\right]$$

*is the Fisher information matrix of $X_t$. At $t = 0$, the identity holds only under the assumption that $\boldsymbol{J}(X)$ exists.*

**Lemma C.3** (Fisher Information Inequality, Rioul (2010)). *Let $X$ and $Y$ be independent random vectors in $\mathbb{R}^d$ with non-zero finite Fisher information $J(X)$ and $J(Y)$. Then the Fisher information of their sum satisfies*

$$J(X + Y)^{-1} \geq J(X)^{-1} + J(Y)^{-1}.$$

**Theorem C.4.** *Let $P$, $P_\theta$, and $Q$ be $d$-dimensional continuous probability distributions on $\mathbb{R}^d$. Let*

$$X \sim P, \quad Y \sim Q, \quad Z \sim \mathcal{N}(0, \sigma_P^2 I_d), \quad Z' \sim \mathcal{N}(0, \sigma_Q^2 I_d),$$

*and define*

$$X + Z \sim P', \quad Y + Z' \sim Q'.$$

*Assume the Fisher information of $X$ and $Y$ are non-zero finite. Then, $\frac{\partial \Delta_E}{\partial \sigma_P^2} < 0$ and $\frac{\partial \Delta_E}{\partial \sigma_Q^2} > 0$*

*Proof.* Let $X_{\sigma_P^2} = X + \sigma_P Z_1 = X + Z$ and $Y_{\sigma_Q^2} = Y + \sigma_Q Z_2 = Y + Z'$ such that $Z_1, Z_2 \sim \mathcal{N}(0, I_d)$. By Lemma C.2, we can derive follows:

$$\frac{d}{d\sigma_P^2}\mathbb{H}(X + Z) = \frac{1}{2}\text{tr}(\mathbf{J}(X + Z)) = \frac{1}{2}J(X + Z)$$

$$\frac{d}{d\sigma_Q^2}\mathbb{H}(Y + Z') = \frac{1}{2}\text{tr}(\mathbf{J}(Y + Z')) = \frac{1}{2}J(Y + Z') \tag{22}$$

where $\mathbf{J}(\cdot)$ is Fisher information matrix.

Then, we can derive follows using Lemma C.3:

$$J(X + Z)^{-1} \geq J(X)^{-1} + J(Z)^{-1} = \frac{1}{J(Z)} \left( \frac{J(X) + J(Z)}{J(X)} \right) = \frac{\sigma_P^2}{d} \left( \frac{J(X) + J(Z)}{J(X)} \right)$$

$$J(Y + Z')^{-1} \geq J(Y)^{-1} + J(Z')^{-1} = \frac{1}{J(Z')} \left( \frac{J(Y) + J(Z')}{J(Y)} \right) = \frac{\sigma_Q^2}{d} \left( \frac{J(Y) + J(Z')}{J(Y)} \right)$$

$$\therefore J(X + Z) \leq \frac{d}{\sigma_P^2} \left( \frac{J(X)}{J(X) + J(Z)} \right), \quad J(Y + Z') \leq \frac{d}{\sigma_Q^2} \left( \frac{J(Y)}{J(Y) + J(Z')} \right)$$

$$(23)$$

By differentiating $\Delta_E$ with respect to $\sigma_P^2$, we obtain:

$$
\begin{aligned}
\frac{\partial \Delta_E}{\partial \sigma_P^2} &= \frac{\partial \left( \frac{d}{2} \log \frac{\sigma_Q^2}{\sigma_P^2} - (\mathbb{H}(Y + Z') - \mathbb{H}(X + Z)) \right)}{\partial \sigma_P^2} \\
&= -\frac{d}{2\sigma_P^2} + \frac{\partial(\mathbb{H}(X + Z))}{\partial \sigma_P^2} \quad \left( \because \frac{\partial(\mathbb{H}(Y + Z'))}{\partial \sigma_P^2} = 0 \right) \\
&= -\frac{d}{2\sigma_P^2} + \frac{J(X + Z)}{2} \quad (\because \text{Equation 22}) \\
&\leq -\frac{d}{2\sigma_P^2} + \frac{d}{2\sigma_P^2} \left( \frac{J(X)}{J(X) + J(Z)} \right) \quad (\because \text{Equation 23}) \\
&= -\frac{d}{2\sigma_P^2} \left( \frac{J(Z)}{J(X) + J(Z)} \right) \\
&< 0
\end{aligned}
$$

$$(24)$$

Similarly, by differentiating $\Delta_E$ with respect to $\sigma_Q^2$, we obtain:

$$
\begin{aligned}
\frac{\partial \Delta_E}{\partial \sigma_Q^2} &= \frac{\partial \left( \frac{d}{2} \log \frac{\sigma_Q^2}{\sigma_P^2} - (\mathbb{H}(Y + Z') - \mathbb{H}(X + Z)) \right)}{\partial \sigma_Q^2} \\
&= \frac{d}{2\sigma_Q^2} - \frac{\partial(\mathbb{H}(Y + Z'))}{\partial \sigma_Q^2} \quad \left( \because \frac{\partial(\mathbb{H}(X + Z))}{\partial \sigma_Q^2} = 0 \right) \\
&= \frac{d}{2\sigma_Q^2} - \frac{J(Y + Z')}{2} \quad (\because \text{Equation 22}) \\
&\geq \frac{d}{2\sigma_Q^2} - \frac{d}{2\sigma_Q^2} \left( \frac{J(Y)}{J(Y) + J(Z')} \right) \quad (\because \text{Equation 23}) \\
&= \frac{d}{2\sigma_Q^2} \left( \frac{J(Z')}{J(Y) + J(Z')} \right) \\
&> 0
\end{aligned}
$$

$$(25)$$

$$\square$$

Therefore, by Theorems C.1 and C.4, we establish that $\Delta_E$ not only becomes strictly positive once the noise variance ratio $\sigma_Q^2/\sigma_P^2$ exceeds a certain threshold, but also exhibits opposite monotonic behaviors with respect to the in/out-of-distribution noise variances. Specifically, $\Delta_E$ exhibits a negative gradient with respect to $\sigma_P^2$ and a positive gradient with respect to $\sigma_Q^2$. Consequently, by decreasing $\sigma_P^2$ while simultaneously increasing $\sigma_Q^2$ such that their ratio surpasses the identified threshold, one can guarantee that $\Delta_E$, as a constituent component of $\Delta$, remains positive and strictly increases.

Since the overall increment $\Delta$ is influenced by $\Delta_{\mathrm{KL}}$, an analysis of $\Delta_{\mathrm{KL}}$ is required in order to explain the expected log-likelihood under both SPEM-noise and SPEM. However, unlike entropy, $\Delta_{\mathrm{KL}}$ is difficult to bound due to the arbitrariness of the OOD setting. Therefore, we first derive in Theorem C.11 a two-sided bound on $\Delta$ under the assumption that $\log P_\theta$ has the $L$-Lipschitz property, in order to analyze the bound of the overall difference $\Delta$.

**Definition C.5** (p-Wasserstein Distance). *Let $P$, $Q$ be probability measures on $\mathbb{R}^d$. Then, p-Wasserstein distance between $P$ and $Q$ is*

$$W_p(P,Q) := \inf_{\gamma \in \Pi(P,Q)} \left( \int ||x-y||^p d\gamma(x,y) \right)^{\frac{1}{p}}.$$

**Lemma C.6** (Wasserstein Distance Inequality, Santambrogio (2015)). *Let $P$, $Q$ be probability measures on $\mathbb{R}^d$ with finite $p$-th moments. Then,*

$$W_1(P,Q) \leq W_p(P,Q), \ \ s.t \ p > 1.$$

**Lemma C.7** (Triangle Inequality of Wasserstein Distance, Santambrogio (2015)). *Let $P$, $Q$, $R$ be probability measures on $\mathbb{R}^d$ with finite $p$-th moments. Then,*

$$W_p(P,R) \leq W_p(P,Q) + W_p(Q,R) \ \ for \ p \geq 1.$$

**Lemma C.8** (Wasserstein Distance between Gaussians, Givens & Shortt (1984)). *Let $Z_1 \sim \mathcal{N}(0,\sigma_1^2 I_d)$, $Z_2 \sim \mathcal{N}(0,\sigma_2^2 I_d)$. Then,*

$$W_2(Z_1,Z_2) = \sqrt{d}|\sigma_1 - \sigma_2|.$$

**Lemma C.9** (Convolution of Wasserstein Distance). *Let $P$, $Q$, $Z$ be probability measures on $\mathbb{R}^d$ with finite $p$-th moments. Then,*

$$W_p(P * Z, Q * Z) \leq W_p(P,Q) \ \ for \ p \geq 1.$$

*Proof.* Let $\gamma^* \in \Pi(P,Q)$ is optimal coupling of $W_p(P,Q)$ and $\Pi(P,Q)$ is coupling of $P$ and $Q$. Then,

$$W_p(P,Q)^p = \int ||x-y||^p d\gamma^*(x,y) \tag{26}$$

Let $T : \mathbb{R}^d \times \mathbb{R}^d \times \mathbb{R}^d \to \mathbb{R}^d \times \mathbb{R}^d$ such that $T(\mathbf{x},\mathbf{y},\mathbf{z}) = (\mathbf{x}+\mathbf{z},\mathbf{y}+\mathbf{z}) = (\mathbf{u},\mathbf{v})$ and $\tilde{\gamma} := T_\#(\gamma^* \otimes Z)$ pushforward of $\gamma^* \otimes Z$ by $T$. Since $U \sim P * Z$ and $V \sim Q * Z$, it follows that $\tilde{\gamma} \in \Pi(P * Z, Q * Z)$. Consequently,

$$
\begin{aligned}
&\int ||\mathbf{u}-\mathbf{v}||^p d\tilde{\gamma} \\
&= \int ||(\mathbf{x}+\mathbf{z})-(\mathbf{y}+\mathbf{z})||^p d(\gamma^* \otimes Z) \\
&= \int ||\mathbf{x}-\mathbf{y}||^p d\gamma^* \\
&= W_p(P,Q)^p
\end{aligned}
\tag{27}
$$

Taking the infimum on both sides, we can derive follows:

$$\inf_{\tilde{\gamma} \in \Pi(P*Z,Q*Z)} \int ||\mathbf{u}-\mathbf{v}||^p d\tilde{\gamma} = W_p(P*Z,Q*Z)^p \leq W_p(P,Q)^p \tag{28}$$

$\square$

**Lemma C.10** ($L$-Lipschitz Wasserstein Distance Inequality). *Let $P$, $Q$ be probability measures on $\mathbb{R}^d$ with finite first moments. Let $f : \mathbb{R}^d \to \mathbb{R}$ be $L$-Lipschitz. Then,*

$$|\mathbb{E}_P[f] - \mathbb{E}_Q[f]| \leq L W_1(P,Q)$$

*Proof.* Because $f$ is $L$-Lipschitz, we can derive follows:

$$
\begin{aligned}
|\mathbb{E}_P[f] - \mathbb{E}_Q[f]| &= \left| \int f(\mathbf{x})dP(\mathbf{x}) - \int f(\mathbf{y})dQ(\mathbf{y}) \right| \\
&= \left| \int f(\mathbf{x}) - f(\mathbf{y})d\gamma(\mathbf{x}, \mathbf{y}) \right| \\
&\leq \int |f(\mathbf{x}) - f(\mathbf{y})|\, d\gamma(\mathbf{x}, \mathbf{y}) \\
&\leq L \int ||\mathbf{x} - \mathbf{y}||d\gamma(\mathbf{x}, \mathbf{y})
\end{aligned}
\tag{29}
$$

where $\gamma$ is arbitrary coupling of $P$ and $Q$. Taking the infimum on both sides, we can derive follows:

$$
\begin{aligned}
|\mathbb{E}_P[f] - \mathbb{E}_Q[f]| &\leq \inf_{\gamma \in \Pi(P,Q)} L \int ||\mathbf{x} - \mathbf{y}||d\gamma(\mathbf{x}, \mathbf{y}) \\
&= LW_1(P, Q)
\end{aligned}
\tag{30}
$$

$\square$

**Theorem C.11.** *Let $P$, $P_\theta$, and $Q$ be $d$-dimensional continuous probability distributions on $\mathbb{R}^d$ and have finite first and second moments. Let*

$$
X \sim P, \quad Y \sim Q, \quad Z \sim \mathcal{N}(0, \sigma_P^2 I_d), \quad Z' \sim \mathcal{N}(0, \sigma_Q^2 I_d),
$$

*and define*

$$
X + Z \sim P', \quad Y + Z' \sim Q'.
$$

*Let $f(\mathbf{x}) = \log P_\theta(\mathbf{x})$ be $L$-Lipschitz. Then*

$$
|\Delta| \leq L(W_2(P, Q) + 2\sqrt{d}|\sigma_Q - \sigma_P|)
$$

*Proof.* By definition of $\Delta$,

$$
\Delta = \mathbb{E}_{\mathbf{x}\sim Z}[\log P_\theta(\mathbf{x})] - \mathbb{E}_{\mathbf{x}\sim Z'}[\log P_\theta(\mathbf{x})] - (\mathbb{E}_{\mathbf{x}\sim P'}[\log P_\theta(\mathbf{x})] - \mathbb{E}_{\mathbf{x}\sim Q'}[\log P_\theta(\mathbf{x})]) \tag{31}
$$

Because of triangle inequality, we can derive follows:

$$
\begin{aligned}
|\Delta| &= |\mathbb{E}_{\mathbf{x}\sim Z}[\log P_\theta(\mathbf{x})] - \mathbb{E}_{\mathbf{x}\sim Z'}[\log P_\theta(\mathbf{x})] - (\mathbb{E}_{\mathbf{x}\sim P'}[\log P_\theta(\mathbf{x})] - \mathbb{E}_{\mathbf{x}\sim Q'}[\log P_\theta(\mathbf{x})])| \\
&\leq |\mathbb{E}_{\mathbf{x}\sim Z}[\log P_\theta(\mathbf{x})] - \mathbb{E}_{\mathbf{x}\sim Z'}[\log P_\theta(\mathbf{x})]| + |\mathbb{E}_{\mathbf{x}\sim P'}[\log P_\theta(\mathbf{x})] - \mathbb{E}_{\mathbf{x}\sim Q'}[\log P_\theta(\mathbf{x})]|
\end{aligned}
\tag{32}
$$

From above equation, we obtain the following:

$$
\begin{aligned}
|\Delta| &\leq |\mathbb{E}_{\mathbf{x}\sim Z}[\log P_\theta(\mathbf{x})] - \mathbb{E}_{\mathbf{x}\sim Z'}[\log P_\theta(\mathbf{x})]| + |\mathbb{E}_{\mathbf{x}\sim P'}[\log P_\theta(\mathbf{x})] - \mathbb{E}_{\mathbf{x}\sim Q'}[\log P_\theta(\mathbf{x})]| \\
&\leq LW_1(Z, Z') + LW_1(P', Q') \ (\because \text{Lemma } C.10) \\
&\leq LW_1(Z, Z') + LW_1(P', Q * Z) + LW_1(Q * Z, Q')(\because \text{Lemma } C.7) \\
&\leq LW_1(Z, Z') + LW_1(P', Q * Z) + LW_1(Z, Z')(\because \text{Lemma } C.9) \\
&\leq 2LW_1(Z, Z') + LW_1(P, Q)(\because \text{Lemma } C.9) \\
&\leq 2LW_2(Z, Z') + LW_2(P, Q)(\because \text{Lemma } C.6) \\
&\leq L(2\sqrt{d}|\sigma_Q - \sigma_P| + W_2(P, Q))(\because \text{Lemma } C.8)
\end{aligned}
\tag{33}
$$

$\square$

By Theorem C.11, we have shown that when $\log P_\theta$ is $L$-Lipschitz, $\Delta$ admits a two-sided bound that consists of the Lipschitz constant, the Wasserstein distance between $P$ and $Q$, and the difference in noise intensity. Although the bound on $\Delta$ includes positive values, it does not explicitly provide conditions under which $\Delta$ becomes strictly positive. Moreover, for analytical convenience, we assumed that the log-likelihood is $L$-Lipschitz; however, this assumption deviates significantly from practical distributions. Therefore, we relax this condition and, in Theorem C.14, derive the bound under the more realistic assumptions that the negative log-likelihood is $L$-smooth and $\lambda$-semiconvex. These conditions can accommodate probability distributions that are weakly multi-modal and provide a more practical setting compared to the $L$-Lipschitz assumption.

**Definition C.12** ($\lambda$-semiconvex function, Payne & Redaelli (2023)). *Let $f : \mathbb{R}^d \to \mathbb{R}$ be a differentiable function. We say that $f$ is $\lambda$-semiconvex for some $\lambda \geq 0$ if the function*

$$\mathbf{x} \; \mapsto \; f(\mathbf{x}) + \frac{\lambda}{2}\|\mathbf{x}\|^2$$

*is convex. Equivalently, for all $\mathbf{x}, \mathbf{y} \in \mathbb{R}^d$,*

$$f(\mathbf{y}) \; \geq \; f(\mathbf{x}) + \langle \nabla f(\mathbf{x}), \mathbf{y} - \mathbf{x} \rangle - \frac{\lambda}{2}\|\mathbf{y} - \mathbf{x}\|^2.$$

**Definition C.13** ($L$-smooth function). *Let $f : \mathbb{R}^d \to \mathbb{R}$ be a differentiable function. We say that $f$ is $L$-smooth for some $L > 0$ if, for all $\mathbf{x}, \mathbf{y} \in \mathbb{R}^d$,*

$$\|\nabla f(\mathbf{x}) - \nabla f(\mathbf{y})\| \; \leq \; L\|\mathbf{x} - \mathbf{y}\|.$$

*Equivalently, $f$ is $L$-smooth if and only if*

$$f(\mathbf{y}) \; \leq \; f(\mathbf{x}) + \langle \nabla f(\mathbf{x}), \mathbf{y} - \mathbf{x} \rangle + \frac{L}{2}\|\mathbf{y} - \mathbf{x}\|^2, \quad \forall \mathbf{x}, \mathbf{y} \in \mathbb{R}^d.$$

**Theorem C.14.** *Let $P$, $P_\theta$, and $Q$ be $d$-dimensional continuous probability distributions on $\mathbb{R}^d$ and have finite first and second moments. Let*

$$X \sim P, \quad Y \sim Q, \quad Z \sim \mathcal{N}(0, \sigma_P^2 I_d), \quad Z' \sim \mathcal{N}(0, \sigma_Q^2 I_d),$$

*and define*

$$X + Z \sim P', \quad Y + Z' \sim Q'.$$

*Let $f(\mathbf{x}) = -\log P_\theta(\mathbf{x})$ be $\lambda$-semiconvex and $L$-smooth. Then*

$$\Delta \in [-C - \frac{d(\lambda + L)}{2}(\sigma_P^2 + \sigma_Q^2), -C + \frac{d(\lambda + L)}{2}(\sigma_P^2 + \sigma_Q^2)]$$

*where $C = \mathbb{E}_{\mathbf{x} \sim P}[\log P_\theta(\mathbf{x})] - \mathbb{E}_{\mathbf{x} \sim Q}[\log P_\theta(\mathbf{x})]$. Also. sufficient condition of $\Delta > 0$ is $-\frac{2C}{d(\lambda + L)} > \sigma_P^2 + \sigma_Q^2$.*

*Proof.* Then, by definition of $\lambda$-semiconvex and $L$-smooth, we can derive following:

$$f(\mathbf{x}) + \langle \nabla f(\mathbf{x}), \mathbf{y} - \mathbf{x} \rangle - \frac{\lambda}{2}\|\mathbf{y} - \mathbf{x}\|^2 \leq f(\mathbf{y}) \; \leq \; f(\mathbf{x}) + \langle \nabla f(\mathbf{x}), \mathbf{y} - \mathbf{x} \rangle + \frac{L}{2}\|\mathbf{y} - \mathbf{x}\|^2$$

$$\Rightarrow \langle \nabla f(\mathbf{x}), \mathbf{y} - \mathbf{x} \rangle - \frac{\lambda}{2}\|\mathbf{y} - \mathbf{x}\|^2 \leq f(\mathbf{y}) - f(\mathbf{x}) \leq \langle \nabla f(\mathbf{x}), \mathbf{y} - \mathbf{x} \rangle + \frac{L}{2}\|\mathbf{y} - \mathbf{x}\|^2 \tag{34}$$

$$\Rightarrow \langle \nabla f(\mathbf{0}), \mathbf{y} \rangle - \frac{\lambda}{2}\|\mathbf{y}\|^2 \leq f(\mathbf{y}) - f(\mathbf{0}) \leq \langle \nabla f(\mathbf{0}), \mathbf{y} \rangle + \frac{L}{2}\|\mathbf{y}\|^2$$

By plugging $Z$ into $\mathbf{y}$, we can derive the following:

$$\mathbb{E}[\langle \nabla f(\mathbf{0}), Z \rangle] - \frac{\lambda}{2}\mathbb{E}[\|Z\|^2] \leq \mathbb{E}[f(Z)] - f(\mathbf{0}) \leq \mathbb{E}[\langle \nabla f(\mathbf{0}), Z \rangle] + \frac{L}{2}\mathbb{E}[\|Z\|^2]$$

$$\Rightarrow -\frac{\lambda}{2}\mathbb{E}[\|Z\|^2] \leq \mathbb{E}[f(Z)] - f(\mathbf{0}) \leq +\frac{L}{2}\mathbb{E}[\|Z\|^2] \tag{35}$$

Then, by applying the same procedure to $Z'$ and expanding difference between the resulting inequalities, we obtain the following:

$$-\frac{\lambda}{2}\mathbb{E}[\|Z\|^2] - \frac{L}{2}\mathbb{E}[\|Z'\|^2] \leq \mathbb{E}[f(Z)] - \mathbb{E}[f(Z')] \leq +\frac{L}{2}\mathbb{E}[\|Z\|^2] + \frac{\lambda}{2}\mathbb{E}[\|Z'\|^2]$$

$$\Rightarrow -\frac{\lambda d}{2}\sigma_P^2 - \frac{L d}{2}\sigma_Q^2 \leq \mathbb{E}[f(Z)] - \mathbb{E}[f(Z')] \leq +\frac{L d}{2}\sigma_P^2 + \frac{\lambda d}{2}\sigma_Q^2$$

$$\Rightarrow -\frac{\lambda d}{2}\sigma_P^2 - \frac{L d}{2}\sigma_Q^2 \leq \mathbb{E}[-\log P_\theta(Z)] - \mathbb{E}[-\log P_\theta(Z')] \leq +\frac{L d}{2}\sigma_P^2 + \frac{\lambda d}{2}\sigma_Q^2 \tag{36}$$

$$\Rightarrow -\frac{L d}{2}\sigma_P^2 - \frac{\lambda d}{2}\sigma_Q^2 \leq \mathbb{E}[\log P_\theta(Z)] - \mathbb{E}[\log P_\theta(Z')] \leq +\frac{\lambda d}{2}\sigma_P^2 + \frac{L d}{2}\sigma_Q^2$$

Meanwhile, the log-likelihood expectations under $P'$ and $Q'$ can be expressed as follows:

$$\begin{aligned}
\mathbb{E}_{\mathbf{x}\sim P'}[\log P_\theta(\mathbf{x})] &= \mathbb{E}_{\mathbf{x}\sim P}[\log P_\theta(\mathbf{x})] + (\mathbb{E}_{\mathbf{x}\sim P'}[\log P_\theta(\mathbf{x})] - \mathbb{E}_{\mathbf{x}\sim P}[\log P_\theta(\mathbf{x})]) \\
\mathbb{E}_{\mathbf{x}\sim Q'}[\log P_\theta(\mathbf{x})] &= \mathbb{E}_{\mathbf{x}\sim Q}[\log P_\theta(\mathbf{x})] + (\mathbb{E}_{\mathbf{x}\sim Q'}[\log P_\theta(\mathbf{x})] - \mathbb{E}_{\mathbf{x}\sim Q}[\log P_\theta(\mathbf{x})])
\end{aligned} \tag{37}$$

Then, by applying the same procedure to $\mathbb{E}_{\mathbf{x}\sim P'}[-\log P_\theta(\mathbf{x})] - \mathbb{E}_{\mathbf{x}\sim P}[-\log P_\theta(\mathbf{x})]$ as was done to obtain the two-sided bounds for log-likelihood expectation difference between $Z$ and $Z'$, we can derive the following:

$$\mathbb{E}[\langle \nabla f(X), Z \rangle] - \frac{\lambda}{2}\mathbb{E}[\|Z\|^2] \leq \mathbb{E}[f(X+Z)] - \mathbb{E}[f(X)] \leq \mathbb{E}[\langle \nabla f(X), Z \rangle] + \frac{L}{2}\mathbb{E}[\|Z\|^2]$$

$$\Rightarrow -\frac{\lambda}{2}\mathbb{E}[\|Z\|^2] \leq \mathbb{E}[f(X+Z)] - \mathbb{E}[f(X)] \leq +\frac{L}{2}\mathbb{E}[\|Z\|^2] \tag{38}$$

$$\Rightarrow -\frac{\lambda d}{2}\sigma_P^2 \leq \mathbb{E}[f(X+Z)] - \mathbb{E}[f(X)] \leq +\frac{Ld}{2}\sigma_P^2$$

Moreover, the following inequality holds for $\mathbb{E}_{\mathbf{x}\sim Q'}[-\log P_\theta(\mathbf{x})] - \mathbb{E}_{\mathbf{x}\sim Q}[-\log P_\theta(\mathbf{x})]$ as well

$$-\frac{\lambda d}{2}\sigma_Q^2 \leq \mathbb{E}[f(Y+Z')] - \mathbb{E}[f(Y)] \leq +\frac{Ld}{2}\sigma_Q^2 \tag{39}$$

Thus, we obtain the following two-sided bound:

$$-\frac{\lambda d}{2}\sigma_P^2 - \frac{Ld}{2}\sigma_Q^2 \leq \mathbb{E}[f(X+Z)] - \mathbb{E}[f(X)] - (\mathbb{E}[f(Y+Z')] - \mathbb{E}[f(Y)]) \leq +\frac{\lambda d}{2}\sigma_Q^2 + \frac{Ld}{2}\sigma_P^2 \tag{40}$$

Because Equation 40 contains a negative sign in the log-likelihood introduced by $f(\cdot)$, adding it to Equation 36 leads to the following inequality:

$$-C - \frac{d(\lambda+L)}{2}(\sigma_P^2 + \sigma_Q^2) \leq \Delta \leq -C + \frac{d(\lambda+L)}{2}(\sigma_P^2 + \sigma_Q^2) \tag{41}$$

where $C = \mathbb{E}_{\mathbf{x}\sim P}[\log P_\theta(\mathbf{x})] - \mathbb{E}_{\mathbf{x}\sim Q}[\log P_\theta(\mathbf{x})]$. Additionally, $-\frac{2C}{d(\lambda+L)} > \sigma_P^2 + \sigma_Q^2$ is sufficient to guarantee $\Delta > 0$.

$\square$

By Theorem C.14, we establish the existence of a guaranteed lower bound when the negative log-likelihood satisfies the $L$-smoothness and $\lambda$-semiconvexity conditions. This existence result holds when $C$ is negative, that is, when the log-likelihood expectation of the OOD exceeds that of the in-distribution. This implies that there exist cases in which SPEM-noise, which does not directly incorporate the original data as input, attains a higher expected log-likelihood difference than SPEM. Consequently, this suggests that SPEM-noise may achieve superior OOD detection performance compared to SPEM in certain case. For future work, it would be of interest to relax the assumption and investigate the effectiveness of SPEM-noise under weaker conditions, and to analyze the conditions under which $\Delta_{KL}$ becomes positive or takes values smaller than $\Delta_E$. Such analyses would further explain when and why SPEM-noise can yield improved OOD detection performance.

## C.6 INFERENCE TIME OF SPEM

SPEM involves two auxiliary steps: extracting embedding vectors using a pretrained model and computing cosine similarity with the in-distribution embedding vectors stored in the memory bank. Let $d$ denote the feature dimension of the extractor and $n$ the number of embedding vectors in the memory bank. Then, the time complexity of this auxiliary process is $\mathcal{O}(nd)$, i.e., linear in both $n$ and $d$. For example, on CIFAR-10, the auxiliary process requires roughly 1 minute of computation about 60,000 train in-distribution images. After this step, the inference time is equivalent to that of methods relying solely on likelihood. Consequently, our method does not introduce substantial additional overhead at inference compared to other approaches.

# D  DESCRIPTION OF LLM USAGE

During the preparation of this manuscript, we used large language models (LLMs) to assist with polishing the writing and surveying related work.

