# OpenReview forum: "Likelihood Paradox Mitigation using Entropy Manipulation with Normalizing Flow in OOD Detection"
_ICLR.cc/2026/Conference — Submitted to ICLR 2026_

### Official Review · Reviewer_WMtm · 2025-10-30

**Soundness:** 3
**Presentation:** 2
**Contribution:** 2
**Rating:** 2
**Confidence:** 3

**Summary:**

This paper investigates the likelihood paradox in normalizing flow–based out-of-distribution (OOD) detection, where OOD samples can receive higher likelihoods than in-distribution (ID) data. The authors propose Semantic Proportional Entropy Manipulation (SPEM), a training-free post-hoc method that perturbs test inputs with Gaussian noise scaled by semantic similarity to ID embeddings. Theoretical results show that entropy manipulation enlarges the expected log-likelihood gap between ID and OOD, and experiments across multiple datasets and models show consistent AUROC improvements.

**Strengths:**

1. The work develops rigorous information theory based on the entropy power inequality and KL-divergence decomposition.  The mathematical reasoning provides valuable insight into the behavior of normalizing flow likelihoods.

2. SPEM operates purely as a post-hoc procedure without retraining the model, requiring only a pretrained encoder and a simple similarity-based noise scaling. This makes it straightforward to implement and integrate into existing normalizing flow OOD detection pipelines.

3. The experiments are extensive, covering different ID/OOD dataset pairs and two representative normalizing flow architectures (ResFlow and Glow). The proposed SPEM and its variant achieve higher AUROC than previous likelihood, ratio, and complexity based methods.

**Weaknesses:**

1. The related work on normalizing flows and the likelihood paradox appears in Section 4, after the proposed method and theoretical analysis. If the readers are not familiar with normalizing flows and the likelihood paradox, these background explanations should appear before the method section to provide necessary context and make the paper more clear.

2. The central idea of this manuscript, modifying the likelihood of normalizing flows through Gaussian noise, has already been investigated in [1], which analyzed the relationship between input complexity and OOD detection performance. Although this manuscript introduces a semantic similarity–based scaling scheme and provides theoretical analysis, the novelty remains limited and should be more clearly distinguished from existing literature.

3. Although the theoretical results are valuable, the empirical evidence does not clearly demonstrate how the observed improvements relate to the proposed theoretical framework. Stronger ablation studies or direct empirical validation linking the theoretical predictions to observed performance would make the paper more convincing.

[1] Osada G., Takahashi T., Nishide T. *Understanding likelihood of normalizing flow and image complexity through the lens of out-of-distribution detection.* AAAI 2024.

**Questions:**

What is the computational cost of maintaining and searching the memory bank during inference compared to other OOD detection methods?

---

> ### Author Response · Authors · 2025-11-15
>
> > **Answer of W1**
>
> We thank the reviewer for pointing out the need for clearer contextualization of the related work and theoretical background. To address this concern, we have revised the manuscript to present the background on normalizing flows and the likelihood paradox earlier to the early part of the manuscript. This improves the logical flow and makes it easier to see how our work builds on and extends prior studies that noted paradoxical likelihood assignments.
>
> > **Answer of W2**
>
> We thank the reviewer for pointing out the relation to [4]. Earlier works [2] and [4] show that normalizing flows tend to assign higher likelihoods to simpler inputs and lower likelihoods to more complex images. Reference [4] also provides a detailed theoretical explanation of this phenomenon from a complexity perspective. In contrast, our work is motivated by the entropic analysis of [6]. We use the decomposition of the expected log likelihood difference into a KL term and an entropy term and ask whether one can actively manipulate the entropy of test inputs, in a similarity-aware manner, so that the likelihood ordering becomes more consistent with human intuition. Concretely, we apply a similarity-dependent transformation to the inputs to control the entropy of the induced distribution and analyze how this affects the ID–OOD log-likelihood gap. Thus, while prior work mainly reports and explains how likelihood varies with image complexity, our contribution is to recast the likelihood paradox in terms of entropy/KL-divergence and to propose a training-free, similarity-based entropy manipulation procedure that reshapes the likelihoods produced by a fixed density model.
>
> > **Answer of W3**
>
> We appreciate the reviewer’s comment regarding the connection between our theoretical and empirical results. As shown in Theorem 2.1, we theoretically demonstrate that detection performance can improve as the entropy of the OOD distribution, $\mathbb{H}(Q)$, increases under perturbation, which is consistent with our empirical finding in Observation 1. Furthermore, Theorem 3.1 extends this result to the setting where perturbations are applied to both in- and out-of-distribution samples, showing that detection performance improves when the perturbation scale is chosen so that it increases $\mathbb{H}(Q)$ more than $\mathbb{H}(P)$. This provides a theoretical explanation for why SPEM works effectively in practice. Empirically, Table 1 (Main Experiment) supports Theorem 3.1 by showing that SPEM consistently outperforms other baselines across various datasets and settings. We hope this clarifies the theoretical–empirical correspondence underlying our method and addresses the reviewer’s concern.

---

> ### Author Response · Authors · 2025-11-15
>
> > **Answer of Q1**
>
> We appreciate the reviewer’s question regarding the computational cost of SPEM. As discussed in Appendix C.6, the additional computation in SPEM consists of two parts:
>
> - Extracting feature embeddings for the training data
> - Computing the maximum cosine similarity between each test embedding and the stored embeddings.
>
> The overall computational complexity of this procedure increases linearly with the number of samples $n$ and the embedding dimension $d$, i.e., $\mathcal{O}(nd)$.
>
> For example, when CIFAR-10 is used as the in-distribution dataset, the memory bank contains 60,000 embedding vectors. In this case, feature extraction took approximately one minute on an RTX 4090Ti GPU, while the subsequent cosine similarity computation incurred negligible overhead. Consequently, the second part of SPEM’s runtime is comparable to that of the likelihood-only method.
>
> We also compared the computational cost of SPEM with other baselines:
>
> - The likelihood ratio method [1] requires training a separate background density model, resulting in a substantially higher computational burden.
> - The typicality method [2] has a similar cost to the likelihood-only method.
> - The complexity method [3] computes PNG-based bit lengths for each data point, which is lightweight.
> - The GMM method [4] incurs moderate cost due to the additional model fitting step in low-dimensional space.
> - The LID method [5] requires Jacobian computation and is therefore significantly more expensive, which is why we used the reported performance from the original paper.
>
> In summary, while SPEM introduces slightly higher computational cost than the likelihood-only, typicality, complexity, and GMM methods due to feature extraction, it remains much more efficient than the likelihood ratio and LID methods.
>
> >**Reference**
>
> [1] Ren, Jie, et al. "Likelihood ratios for out-of-distribution detection." *Advances in neural information processing systems* 32 (2019).
>
> [2] Nalisnick, Eric, et al. "Detecting out-of-distribution inputs to deep generative models using typicality." *arXiv preprint arXiv:1906.02994* (2019).
>
> [3] Serrà, Joan, et al. "Input complexity and out-of-distribution detection with likelihood-based generative models." arXiv preprint arXiv:1909.11480 (2019).
>
> [4] Osada, Genki, Tsubasa Takahashi, and Takashi Nishide. "Understanding likelihood of normalizing flow and image complexity through the lens of out-of-distribution detection." *Proceedings of the AAAI Conference on Artificial Intelligence*. Vol. 38. No. 19. 2024.
>
> [5] Kamkari, Hamidreza, et al. "A geometric explanation of the likelihood OOD detection paradox." *arXiv preprint arXiv:2403.18910* (2024).
>
> [6] Caterini, Anthony L., and Gabriel Loaiza-Ganem. "Entropic issues in likelihood-based ood detection." I (Still) Can't Believe It's Not Better! Workshop at NeurIPS 2021. PMLR, 2022.

---

> ### Author Response · Authors · 2025-11-25
>
> Dear Reviewer WMtm,
>
> We would like to kindly check whether our rebuttal sufficiently clarified your concerns. If any issues remain, we would be glad to provide additional explanation.

---

### Official Review · Reviewer_maRj · 2025-11-01

**Soundness:** 1
**Presentation:** 3
**Contribution:** 1
**Rating:** 2
**Confidence:** 4

**Summary:**

This paper aims to address the "problematic phenomenon" first identified in [1]: when using the likelihood of a probabilistic generative model as the criterion for out-of-distribution (OOD) detection, certain dataset pairs exhibit the unintuitive behavior that OOD samples receive **higher likelihoods** than in-distribution (ID) samples. Formally, this can be written as:

$\mathbb{E}_{x\sim P} [\log P_{\theta} (x)] < \mathbb{E}_{x\sim Q} [\log P_{\theta} (x)]$

where $P$ denotes the distribution of in-distribution (ID) data, $Q$ denotes the OOD data distribution, and $P_\theta$ is the generative model.

[2] provided an entropic explanation for this phenomenon through the following decomposition:

$\mathbb{E}_{x\sim P} [\log P_{\theta} (x)] - \mathbb{E}_{x\sim Q} [\log P_{\theta} (x)] = KL(Q||P_\theta) - KL(P||P_\theta) + \mathbb{H}(Q) - \mathbb{H}(P)$

When the entropy of the in-distribution is higher than that of the OOD data, i.e., $\mathbb{H}(P) > \mathbb{H}(Q)$, the right-hand side can become negative, thereby explaining the counterintuitive result observed in [1].

Building on this entropic perspective, the current paper argues that increasing the entropy of the OOD data $\mathbb{H}(Q)$ via Gaussian perturbations can mitigate and potentially resolve this issue.

To this end, the authors propose the **SPEM (Semantic Proportional Entropy Manipulation)** algorithm, which adaptively adds Gaussian noise to OOD-like inputs to increase their entropy and correct the likelihood ordering between in-distribution and out-of-distribution samples.

**Reference**

[1] Nalisnick, Eric, et al. "Do deep generative models know what they don't know?." arXiv preprint arXiv:1810.09136 (2018).

[2] Caterini, Anthony L., and Gabriel Loaiza-Ganem. "Entropic issues in likelihood-based ood detection." I (Still) Can't Believe It's Not Better! Workshop at NeurIPS 2021. PMLR, 2022.

**Strengths:**

The paper is clearly written and easy to follow. The overall flow, from the problem formulation, to the motivation, and finally to the detailed method description, is well organized and presented in a coherent and understandable manner.

**Weaknesses:**

This paper suffers from a **fundamental conceptual issue**.

First, [2] provides an explanation for the phenomenon described in [1]; it does not define a OOD detection problem itself.

The paper raises the question:

> If we increase OOD entropy via perturbation, wouldn’t the expected log-likelihood difference align with intuition?

However, this is **not a fundamental question** for solving OOD detection. My answer would be: yes, if we artificially increase the entropy of OOD data by any means, the numerical results can indeed be made to align with our intuition. Yet this only adjusts the surface-level metrics to appear more intuitive, it does not actually address the core challenge of detecting out-of-distribution samples. The earlier workshop paper [2] merely interpreted the paradox from an entropic perspective; logically, such an explanation cannot be inverted to construct a genuine solution.

Due to this misunderstanding, the proposed SPEM method ends up being a form of **circular reasoning**. It requires knowing (or at least assuming) that a sample is likely to be OOD in order to assign stronger Gaussian perturbations and thereby increase its entropy. But determining such OOD likelihood is precisely the purpose of an OOD detection method. If another mechanism must first identify OOD tendencies, then the practical meaning of SPEM as an OOD detector becomes questionable.


Because of these fundamental concerns, I lean toward a Reject (Score: 2). However, I acknowledge that the authors have clearly articulated their reasoning and presented their ideas in a coherent manner. I remain open to discussion: if the authors can convincingly clarify or rebut these conceptual issues in their response, I would be willing to reconsider my position.


**Reference**

[1] Nalisnick, Eric, et al. "Do deep generative models know what they don't know?." arXiv preprint arXiv:1810.09136 (2018).

[2] Caterini, Anthony L., and Gabriel Loaiza-Ganem. "Entropic issues in likelihood-based ood detection." I (Still) Can't Believe It's Not Better! Workshop at NeurIPS 2021. PMLR, 2022.

**Questions:**

See **Weaknesses**.

---

> ### Author Response · Authors · 2025-11-15
>
> We appreciate the reviewer’s insightful comments.
>
> Prior work [1] shows that likelihood-based generative models can assign higher likelihood to OOD than to in-distribution (ID) data, and we agree that [2] offers a conceptual rather than a complete solution.
>
> Several methods (e.g., LID [3], PNG complexity [4]) also use label-free auxiliary signals; these are typically treated as analytic corrections, not circular reasoning. SPEM is similar in spirit but operates directly on the entropy to realign the model’s likelihood behavior. We acknowledge that a naive threshold on $ 1-\lambda $ can yield detection performance comparable to SPEM on some pairs; this reflects rank monotonicity, not circularity. Our contribution is to expose and control the density model’s likelihood behavior through entropy manipulation and to base decisions in post-perturbation $\log P_\theta$.
>
> Our goal is to realign likelihood behavior via a training-free, entropy-driven intervention: we ask whether increasing the entropy of OOD-like samples can make the likelihood ordering more consistent with intuition. We provide both theoretical and empirical support for this claim (Theorem 3.1; Table 1). This is not merely changing a score; it addresses a conceptual gap by showing that a trained model’s likelihood can be systematically reshaped without retraining, clarifying how estimated likelihood responds to input concentration and thereby realigning the expected likelihood ordering between ID and OOD.
>
> Within SPEM, the feature-extractor similarity ($ \lambda $) is label-free and used only to scale the perturbation; the decision statistic is the model’s post-perturbation $\log P_\theta$, which our theory links to the expected ordering. Although the score can be used as a detector once a decision rule (e.g., a threshold) is applied, our emphasis is the underlying statistic--the post-perturbation $\log P_\theta$--which we analyze theoretically; the detector is simply an application of this statistic. Notwithstanding this focus, the resulting detector attains superior performance on several benchmark pairs compared to likelihood-based comparison models, as evidenced in Table 1.
>
> Finally, the phrase "to enhance likelihood-based OOD detection" describes the evaluation setting, not our core claim. The contribution is showing that entropy control realigns likelihood assignment without retraining, offering theoretical and empirical insight into the likelihood paradox while showing its superior OOD detection performance relative to comparison baselines.
>
> We hope this clarification resolves the reviewer’s concern.
>
> >**References**
>
> [1] Nalisnick, Eric, et al. "Do deep generative models know what they don't know?." arXiv preprint arXiv:1810.09136 (2018).
>
> [2] Caterini, Anthony L., and Gabriel Loaiza-Ganem. "Entropic issues in likelihood-based ood detection." I (Still) Can't Believe It's Not Better! Workshop at NeurIPS 2021. PMLR, 2022.
>
> [3] Kamkari, Hamidreza, et al. "A Geometric Explanation of the Likelihood OOD Detection Paradox." *International Conference on Machine Learning*. PMLR, 2024.
>
> [4] Serrà, Joan, et al. "Input complexity and out-of-distribution detection with likelihood-based generative models." *arXiv preprint arXiv:1909.11480* (2019).

---

> > ### Comment · Reviewer_maRj · 2025-11-15
> >
> > Thanks for your reply. Here are my comments:
> >
> > > Several methods (e.g., LID [3], PNG complexity [4]) also use label-free auxiliary signals; these are typically treated as analytic corrections, not circular reasoning. SPEM is similar in spirit but operates directly on the entropy to realign the model’s likelihood behavior.
> >
> > I disagree with this characterization. The auxiliary signals used in [3] and [4] (LID and PNG complexity) are **independent of the OOD task**. In contrast, your signal $\lambda$ is itself an OOD criterion: $\lambda$ is the **maximum cosine similarity between the test sample’s features and in-distribution features**. Therefore, in effect, you are using an OOD criterion as an auxiliary signal to penalize likelihood, and then presenting the resulting likelihood behavior as your OOD detection method. This is why I consider your approach to involve **circular reasoning**.
> >
> > > Our goal is to realign likelihood behavior via a training-free, entropy-driven intervention: ...
> >
> > From my perspective, your method essentially treats samples identified as OOD by cosine similarity as inputs to which noise is added to artificially inflate entropy. Since methods that rely on likelihood naturally classify higher-entropy inputs as OOD, the approach feels superficial, which is the same concern I raised previously.
> >
> > > Within SPEM, the feature-extractor similarity ($\lambda$) is label-free and used only to scale the perturbation;
> >
> > This phrasing seems to downplay the role of $\lambda$. Yet in my view, $\lambda$ already has the capacity to detect OOD samples on its own. If that is the case, why not use $\lambda$ directly as the OOD criterion?

---

> > > ### Author Response · Authors · 2025-11-17
> > >
> > > We thank the reviewer for their careful and constructive remarks.
> > >
> > > >**On whether LID is “independent” of OOD detection**
> > >
> > > We respectfully use LID-based OOD detection as an illustrative example. In a typical LID method [1], an auxiliary statistic—the local intrinsic dimension (LID)—is combined with the likelihood from a density model. A test sample is classified as in-distribution only when both its LID and its model likelihood exceed pre-defined thresholds $\phi_{\text{LID}}$ and $\phi_{L}$; all remaining cases are treated as OOD. Concretely, the three OOD cases are:
> > >
> > > - **Likelihood and LID are both below their thresholds**
> > > - **Likelihood is above its threshold but LID is below**
> > > - Likelihood is below its threshold but LID is above
> > >
> > > In cases (1) and (2), the decision effectively depends on LID alone, regardless of the likelihood. Thus the LID statistic is not independent of OOD detection; rather, the method assumes that samples with low LID are more likely to be OOD and uses this assumption directly in the decision rule. In the reviewer’s own words, this “It requires knowing (or **at least assuming**)” Under that interpretation, LID-based detection could also be viewed as a form of “circular reasoning.”
> > >
> > > We do not claim that LID is flawed because of this; instead, our point is that using an auxiliary, label-free statistic that correlates with OOD-like samples is a standard inductive bias in likelihood-based OOD detection. Our method follows the same pattern: we agree that the similarity $\lambda$ plays an important role and serves as a criterion correlated with OOD tendency. The contribution, however, is that we use this criterion only to guide entropy manipulation using only in-distribution training data, thereby changing the density model’s assigned likelihoods and aligning them with human intuition.
> > >
> > > >**On “methods that rely on likelihood naturally classify higher-entropy inputs as OOD”**
> > >
> > > We fully agree that prior work has observed a strong connection between entropy of the data distribution and the likelihood paradox. In particular, [2] argues that the paradox arises because certain OOD datasets have lower entropy than the ID dataset. Our work takes this observation one step further: instead of passively analyzing fixed distributions, we actively change the entropy of the target distribution via perturbations and study how this affects the expected log-likelihood gap.
> > >
> > > We show theoretically that, under Gaussian perturbations, increasing the entropy of the OOD distribution relative to the ID distribution can increase a lower bound on the expected log-likelihood difference (Theorem 3.1). We then validate this effect empirically, demonstrating that our entropy manipulation both mitigates the likelihood paradox and yields improved OOD detection performance compared to baseline methods in our main text.
> > >
> > > If our method were proposed without this theoretical background, we would agree that it might look superficial, focusing only on performance. However, the core of our contribution is precisely the information-theoretic analysis that explains why such entropy manipulation can mitigate the paradox under specific conditions. We therefore emphasize that our focus is not simply on achieving high AUROC, but on understanding how controlled entropy changes reshape likelihood behavior in generative models.

---

> > > > ### Author Response · Authors · 2025-11-17
> > > >
> > > > >**On “why not use $\lambda$ directly as the OOD criterion?”**
> > > >
> > > > We acknowledge that, from a pure performance perspective, using $\lambda$ directly as an OOD score could be a simple and effective alternative. However, the central question of our work is not “what is the best detector” but rather:
> > > >
> > > > ***How can we mitigate the likelihood paradox and understand the behavior of likelihood-based generative models?***
> > > >
> > > > A detector that relies only on $\lambda$ answers a different question. It completely bypasses the generative model and therefore does not shed light on how the model assigns likelihoods, or why OOD samples sometimes receive higher likelihood than ID samples. Such a detector might perform well, but it would not provide any explanation or remedy for the paradoxical likelihood ordering.
> > > >
> > > > We acknowledge that $\lambda$ itself has discriminative capacity, but in SPEM we explicitly use it as a semantic control signal to modulate the perturbation scale; the statistic we focus on theoretically and empirically is the post-perturbation log-likelihood $\log P_\theta$. This is exactly the quantity that our theory analyzes: our main theoretical results (e.g., Theorem 3.1) are expressed in terms of the expected log-likelihood under perturbed distributions. Our analysis characterizes conditions under which this statistic becomes more aligned with the intuitive likelihood ordering.
> > > >
> > > > Therefore, even in our response to Reviewer k3Ki, we emphasized that strong empirical performance alone does not mean that the OOD detection problem based on likelihood-based method has been solved, and we explicitly state this limitation in the conclusion of our paper. Ultimately, we believe that an optimal solution to the “likelihood paradox in OOD detection using density estimation models” will require approaches that directly shape the training process—e.g., through appropriate regularization—so that the learned likelihood function itself aligns with human intuition, rather than relying on auxiliary signals at test time. We regard this as an important direction for future work. We hope that our analysis of entropy-based interventions can serve as a useful stepping stone toward such methods, and we would be grateful if the reviewer could consider our work within this broader perspective.
> > > >
> > > > We hope that our response has clarified the point and addressed the reviewer’s concern.
> > > >
> > > > >**References**
> > > >
> > > > [1] Kamkari, Hamidreza, et al. "A Geometric Explanation of the Likelihood OOD Detection Paradox." *International Conference on Machine Learning*. PMLR, 2024.
> > > >
> > > > [2] Caterini, Anthony L., and Gabriel Loaiza-Ganem. "Entropic issues in likelihood-based ood detection." I (Still) Can't Believe It's Not Better! Workshop at NeurIPS 2021. PMLR, 2022.

---

> ### Author Response · Authors · 2025-11-25
>
> Dear Reviewer maRj,
>
> We would like to kindly check whether our rebuttal sufficiently clarified your concerns. If any issues remain, we would be glad to provide additional explanation.

---

> ### Comment · Reviewer_maRj · 2025-11-26
>
> Thank you for the detailed response. Below are my comments:
>
> ## **On the “independence” of auxiliary statistics**
>
> Allow me to clarify what I meant by **independent**. Attributes such as LID [1], PNG complexity [2], likelihood under a generative model trained on a general natural-image distribution [3], or likelihood under a corruption-trained generative model [4] were **not originally developed for OOD detection**. Only later did subsequent works discover that these attributes correlate with OOD behavior and introduced them as penalty terms for likelihood-based unsupervised OOD detection. A major contribution of those works is precisely that—through intuition and analysis—they uncovered a non-obvious connection between those quantities and the OOD task.
>
> Strictly speaking, you are right that “independent” is not the perfect word. What I meant is that these attributes were **not designed for the OOD task itself**, rather than being completely “independent” or “irrelevant.” I appreciate your pointing this out, but this clarification does not change my overall position.
>
> Regarding why LID constitutes a contribution whereas your $\lambda$ does not: at the very least, LID introduces a new penalty term by uncovering a structural relationship between an intrinsic attribute and OOD detection. In contrast, $\lambda$ in your method is simply a **built-in OOD criterion**—something that is already explicitly indicative of OODness—and using it to modulate perturbation does not constitute a novel contribution.
>
> ## **On circular reasoning**
>
> Let me summarize your method:
>
> > Because high-entropy data receive lower likelihood, you artificially increase the entropy of data you already believe to be OOD via Gaussian noise, so that the likelihood-based detector now identifies them as OOD. In this way, you “fix” the likelihood paradox.
>
> The circularity arises because the data whose entropy you manipulate are exactly those you **already regard as likely OOD**. You decrease their likelihood by injecting noise **because** you have determined (via $\lambda$) that they are OOD-like. This then makes a likelihood-based criterion appear effective.
>
> Is this not circular reasoning?
>
> ## **On the purported “contribution” and its superficiality — entropy manipulation is not addressing the likelihood paradox**
>
> Even ignoring the circularity issue, your claimed contribution is “manipulating entropy.” As I stated in my original review, this is merely a superficial numerical adjustment and does not fundamentally relate to the likelihood paradox. Let me clarify this more explicitly:
>
> * Likelihood paradox: using $p(x)$ as the OOD criterion, some OOD samples receive **higher** likelihood than ID samples.
> * Your manipulation: use $p(x + \lambda (x) \epsilon)$ as the OOD score, where $\epsilon \sim N(0, I)$, and $\lambda(x)$ is itself an OOD criterion based on $x$. Here I make the dependency $\lambda = \lambda(x)$ explicit to emphasize that, although we usually write $\lambda$ for brevity, it is fundamentally a function of the input $x$.
>
> This procedure does nothing more than exploiting the known fact that “high-entropy samples have low likelihood.” The method **pretends** to solve the paradox by altering the input distribution, but it no longer concerns the original paradox of $p(x)$ at all.
>
>
> ## **On SPEM-noise**
>
> A related question: what is the difference between SPEM-noise and directly using $\lambda$ as the OOD score? Does SPEM-noise not serve as evidence that $\lambda$ itself is a better OOD criterion than SPEM? And does this not further suggest that the proposed entropy-manipulation scheme is irrelevant to $p(x)$, and that good OOD detection is achieved solely through $\lambda$?
>
> ## **Summary**
>
> Together with the concerns raised in my original review, I believe the paper reflects a fundamental misunderstanding of what constitutes a solution to the likelihood paradox. The proposed method is ultimately a superficial numerical manipulation that does not resolve the underlying issue, and it relies on circular reasoning: wrapping an existing deterministic OOD measure in a probabilistic formulation to create the appearance of a solution.
>
> I lean toward a **strong reject**, though I am maintaining a score of 2 out of courtesy to the reviewing process.
>
> **References**
>
> [1] Kamkari, Hamidreza, et al. "A Geometric Explanation of the Likelihood OOD Detection Paradox." International Conference on Machine Learning. PMLR, 2024.
>
> [2] Serrà, Joan, et al. "Input complexity and out-of-distribution detection with likelihood-based generative models." arXiv preprint arXiv:1909.11480 (2019).
>
> [3] Schirrmeister, Robin, et al. "Understanding anomaly detection with deep invertible networks through hierarchies of distributions and features." Advances in Neural Information Processing Systems 33 (2020): 21038-21049.
>
> [4] Ren, Jie, et al. "Likelihood ratios for out-of-distribution detection." Advances in neural information processing systems 32 (2019).

---

> ### Author Response · Authors · 2025-12-02
>
> We thank the reviewer for the detailed comments. We address the concerns about (i) superficiality of the entropy manipulation and (ii) circularity / the role of the flow model, and clarify the scope of our contribution.
>
> >**On whether our entropy manipulation is a superficial numerical adjustment.**
>
> We first clarify what we mean by “addressing” the likelihood paradox. In line with the reviewer, we understand the paradox as the phenomenon that some OOD samples obtain higher likelihood than ID samples when one directly uses the likelihood $p_\theta(x)$ of a trained generative model as an OOD score. In this sense, “solving” or “mitigating” the paradox means making the ordering induced by the score more consistent with our semantic intuition about which samples should be considered OOD. Our method does precisely this for a flow model: instead of using $p_\theta(x)$ as the OOD score, we use $p_\theta(x+\lambda(x))$, and we empirically observe that the resulting likelihood ordering is much better aligned with semantic notion of OODness and yields improved detection performance on classic failure cases. Even setting circularity aside for the moment, we therefore believe it is fair to say that we mitigate the paradox at the level of likelihood-based ordering.
>
> Also, the reviewer argues that this effect is merely a consequence of the well-known fact that high-entropy distributions tend to have lower likelihood than low-entropy ones. We agree that this basic relationship is known, but our contribution is not to restate this heuristic. Rather, Theorems 3.1 and 4.1 show that, under our entropy manipulation scheme based on $\lambda(x)$, the expected likelihood ordering between an in-distribution $P$ and an OOD $Q$ can be changed in a controlled way, and can be brought in line with our intuitive OOD ranking under explicit conditions. In other words, we use $\lambda(x)$ to define a concrete transformation $x \mapsto x + \lambda(x)$ and then provide a theoretical explanation of when this transformation will realign the likelihood-based ordering with semantic intuition. For this reason, we do not view our method as a superficial numerical adjustment that simply “turns up entropy”; it is a theoretically motivated test-time procedure whose design and effect are directly supported by our analysis.

---

> ### Author Response · Authors · 2025-12-02
>
> >**On circularity, the role of $\lambda(x)$, and why we use a flow at all.**
>
> We also wish to clarify our perspective on the reviewer’s notion of circularity. Our method never accesses OOD data or labels; the quantity $\lambda(x)$ is a label-free similarity statistic derived from a representation, and the same mapping $x \mapsto x + \lambda(x)$ is applied to all test samples according to a fixed rule. The final OOD score is always the model’s likelihood $p_\theta(x+\lambda(x))$, not $\lambda(x)$ itself. In a strict logical sense, we are therefore not assuming the OOD labels that we later “recover” with our score. Even if one prefers to describe the use of a similarity signal as “circular” in an informal sense, we would like to emphasize that, without any OOD supervision (or exposure), our procedure still consistently mitigates paradoxical behavior of the flow’s likelihood and improves OOD detection over using $p_\theta(x)$ directly. We believe this makes the approach practically meaningful regardless of terminology.
>
> This also explains why we explicitly use a flow model rather than relying on $\lambda(x)$ alone. The goal of our work is not to propose yet another stand-alone discriminative OOD score, but to understand and adjust the behavior of likelihood-based generative models under the likelihood paradox. Using $\lambda(x)$ by itself already shows that the representation carries useful OOD information, but it does not answer the central question raised by the paradox: can the likelihood of a  flow be brought into agreement with a reasonable semantic notion of OODness? By studying $p_\theta(x+\lambda(x))$, we show that a simple, $\lambda(x)$-guided entropy manipulation can realign the flow’s likelihood with such a notion and quantify when this happens. Although the OOD detection performance of a $\lambda$-only detector is comparable to that of using $p_\theta(x+\lambda(x))$, it does not reveal how the likelihood of a fixed flow model can be realigned with a semantic notion of OODness through a simple $\lambda(x)$-guided entropy manipulation.
>
>
> Finally, the SPEM-noise variant complements this perspective by highlighting the role of the original data distribution’s entropy structure. By applying entropy manipulation that is determined solely by $\lambda(x)$, it illustrates how the entropy structure of the original image distribution itself can interfere with likelihood ordering, and invites a closer examination of how much of the flow’s likelihood is determined by semantic content versus low-level distributional properties. Taken together, SPEM and SPEM-noise shed light on how entropy, semantic similarity, and the likelihood of a flow model interact in the context of the likelihood paradox.

---

### Official Review · Reviewer_k3Ki · 2025-11-05

**Soundness:** 3
**Presentation:** 3
**Contribution:** 3
**Rating:** 8
**Confidence:** 4

**Summary:**

The authors consider OOD detection, where a model $P_\theta$ trained on samples from $P$ must decide whether a sample $X \sim Q$ satisfies $Q = P$ or $Q \neq P$. The authors draw on earlier literature showing that counterintuitive results can occur when the entropy of $P$ (the in-distribution) is larger than the entropy of $Q$ (the out-distribution). This leads to cases where $P_\theta$ can assign higher likelihoods to samples from $Q$ than from $P$ when the two differ.

Inspired by this, the authors propose an “entropy manipulation” method based on a nonparametric nearest-neighbor–type idea: all inputs to the OOD test are perturbed by noise, but that noise is scaled higher when $\text{embed}(x_{\text{test}})$ is not cosine-similar to any $\text{embed}(x_i)$ in a stored set of $x_i$ from $P$, using a pretrained semantic embedding model. (The scale depends on the most similar $x_i$.)

Formally, the authors draw on a bound that shows the benefits of increasing the entropy of $P$- and $Q$-input samples, but in a specific way that raises the entropy more for samples from $Q$. This has the effect of increasing the separation between $\mathbb{E}P[\log P\theta]$ and $\mathbb{E}Q[\log P\theta]$, improving OOD detection.

**Strengths:**

- Thorough familiarity with, and citation of, relevant background work

- Clear theorems and intuition on how to “separate” samples from $P$ and $Q$ to improve OOD tests

- A method presented that can sometimes achieve this separation

- Careful and complete quantitative evaluation on many common OOD dataset pairs, with results also reported for existing OOD tests

**Weaknesses:**

Non specifically, but I have some questions, see below.

**Questions:**

**Question about Equation (2)**

“In cases where detection is performed solely by comparing likelihoods, the following inequality holds,

$$E_P[\log P_\theta] < E_Q[\log P_\theta]$$"

What does ‘it holds’ mean here? Do you mean ‘it always holds,’ or ‘surprisingly, it (only) sometimes holds’?

Cross-entropy is greater than or equal to entropy, and these are negatives of those quantities, so negative cross-entropy $<$ negative entropy. The inequality could be true or false depending, for example, on whether $P_\theta = P$ or $P_\theta = Q$.

Could you please clarify what you meant more precisely? It would help to edit the sentence before (2) and to follow (2) with a short explanation.

**Results in Table 1**

There are several OOD accuracies equal to 1 or close to 1 in Table 1. Could you elaborate on what this means, especially relative to the first row (“Likelihood”)?

Does this mean that, formally, the distributions of the test statistic $\log P_\theta$ are disjoint for the two datasets after applying SPEM, while they might have been quite overlapping originally (e.g., leading to $\approx 0.58$ accuracy on CelebA/CIFAR)? Such a large jump seems striking and deserves some discussion.

More generally, many results show your method achieving near-optimal OOD detection on standard datasets in this subarea of machine learning. Could you provide further analysis on what this means? Why or why not do you believe these problems are now “solved”?

I do see in the additional experiment with Gaussian inputs of varying scales that you point out that entropy is not the whole story. How does that observation relate to the results on the real datasets?


**Statistical properties of the optimal embedding model for the scaled Gaussian**

It would be especially interesting to hear more about the following: using the bound in Equation (3) and the results of Theorem 3.1, what can we say about the optimal embedding model that achieves the $\lambda$-scaling for the noise that maximally separates the $P$ and $Q$ cross-entropies with $\log P_\theta$?

What properties must these embeddings have in terms of $P$ and $Q$, and what are some cases where it is or is not achievable to improve over the standard likelihood test using the presented method?

An example property might be something like non-overlap in the densities of the random variables defined by the pushforward through the embedding.

In other words, the fact that we are modifying inputs from P or Q based on their estimated relationship to samples from P, seems to imply that we are augmenting our OOD statistic with another OOD statistic, where the latter statistic may itself be subject to caveats of OOD statistics. What are those?

---

> ### Author Response · Authors · 2025-11-15
>
> We sincerely thank the reviewer for their careful reading of our work and for the constructive comments provided.
>
> >**Question about Equation (2)**
>
> Equation (2) states that the expected log-likelihood estimated by a density estimation model $P_{\theta} $ (e.g., a normalizing flow) for the OOD distribution ($Q$) exceeds that assigned to the in-distribution ($P$). This corresponds to the middle histogram in Figure 1 (CIFAR-10 vs. SVHN). However, the same inequality could also arise in degenerate cases where only a very small subset of OOD samples receives extremely high likelihood values while most OOD samples receive lower likelihoods than in-distribution samples. To avoid such potential misinterpretation, we revise the sentence before Equation (2) to clarify that this inequality does not hold universally but instead describes a phenomenon often observed in practice.
>
> >**Results in Table 1**
>
> The paper focuses on the likelihood reversal phenomenon, where the model assigns higher likelihoods to visually distinct OOD samples than to ID samples. This is evident from the low AUROC values in the first row of Table 1. That SPEM achieves near-perfect AUROC on these pairs shows that adjusting the entropy of two distributions can realign the likelihood assignments with human intuition. As this process makes the test statistic, $\log P_{\theta}$, almost disjoint between in- and out-of-distribution samples, the performance improvement can be interpreted precisely in the way the reviewer has described.
>
> We do not claim that the likelihood-based OOD detection problem has been completely solved. This perspective is also reflected in the conclusion section of the main text. SPEM currently uses a feature extractor pretrained on general image data to control the strength of entropy manipulation. However, we believe that the fundamental paradox in the likelihood assignment of generative models can only be considered truly resolved when the model itself---without any auxiliary process---assigns well-calibrated likelihoods through its own training dynamics. For instance, incorporating a regularization term during training that penalizes high likelihood assignments to samples inferred to come from high-entropy regions. We hope that future research will explore such directions to address this paradox in a more intrinsic manner.
>
> As explained in Section 6 (Sensitivity of Likelihood), we have clarified that entropy alone does not account for the observed sensitivity. It is generally known that the entropy order among image datasets follows SVHN < CIFAR-10 $\approx$ CelebA [1]. This ordering is also consistent with the AUROC values reported in the “Likelihood” row of Table 1. Based on this, we initially expected CIFAR-10 and CelebA to exhibit similar sensitivity patterns to Gaussian perturbations. However, our experiments revealed that CelebA instead behaves more similarly to SVHN. We interpret this as evidence that the divergence term $D_{KL}(Q || P_{\theta})$, which captures the distributional discrepancy between OOD data and the model density, also plays a significant role in shaping this sensitivity. We emphasize that a deeper analysis of the influence of $D_{KL}(Q || P_{\theta})$ remains an important direction for future work.

---

> ### Author Response · Authors · 2025-11-15
>
> >**Statistical properties of the optimal embedding model for the scaled Gaussian**
>
> A model that can extract the optimal embedding for maximizing the separation of $\lambda$ would be one in which embeddings of in-distribution samples exhibit very high similarity with the training in-distribution embeddings, while embeddings of OOD samples show very low similarity. In other words, the optimal embedding model should map in-distribution and OOD samples such that their representations become highly separable in the embedding space. If the embeddings are normalized to lie on a $d$-dimensional unit sphere, this condition implies that in-distribution and OOD embeddings should ideally be positioned on opposite sides of the origin.
>
> The relative performance between SPEM and the standard likelihood test can be expressed by Equation (3). Intuitively, this means that SPEM can outperform the standard likelihood test when the increase in OOD entropy induced by the perturbation exceeds both the increase in in-distribution entropy and the additional divergence caused by the noise term (i.e., the increase in $D_{KL}(Q || P_{\theta})$. Although the KL-divergence term involving $Q$ is difficult to analyze due to the inherent randomness of OOD data, this formulation suggests that as long as the OOD entropy increase is sufficiently large and the in-distribution entropy increase is kept small, superior performance of SPEM can be achieved. Conversely, when this condition is not met—i.e., when the OOD entropy increase is not large enough—Equation (3) becomes negative, corresponding to cases where SPEM fails to surpass the standard likelihood test.
>
> When the likelihoods are already inverted even before applying the perturbation, certain distributional properties of the data can interfere with SPEM’s detection performance. We empirically demonstrate this phenomenon through a variant of our method, SPEM-noise, which estimates likelihoods solely on noise samples generated without using the original images. Interestingly, this variant achieves higher detection performance than the standard SPEM, as theoretically explained in Appendix C.5. We agree with the reviewer’s perspective that the original test statistic can, in some cases, hinder detection. These observations prompt further reflection on whether the original data truly contributes to aligning the likelihood assignment with human intuition.
>
> >**Reference**
>
> [1] Serrà, Joan, et al. "Input complexity and out-of-distribution detection with likelihood-based generative models." *arXiv preprint arXiv:1909.11480* (2019).

---

> ### Author Response · Authors · 2025-11-25
>
> Dear Reviewer k3Ki,
>
> We would like to kindly check whether our rebuttal sufficiently clarified your concerns. If any issues remain, we would be glad to provide additional explanation.

---

### Meta-Review · Area_Chair_trMH · 2025-12-08

**Summary:**

1) Reviewer maRj raised a significant concern that the paper employs "circular reasoning" to achieve its goal. The paper addresses a well-known problem of paradoxical OO-detection results when using likelihood models on datasets of differing complexity. Building on analysis of past work (Caterini & Loaiza-Ganem), the authors propose that adding Gaussian noise to OOD data can correct the anomalous behaviour. The reviewer points out that doing so requires having OOD data labeled as OOD, which is essentially assuming knowledge of the target variable. Further discussion raised comparisons with prior OOD detection methods.

2) Reviewer k3Ki points out an inequality which appears incorrect, asks whether OOD detection is now a solved problem, and asks about an optimal embedding model.

3) Reviewer WMtm references prior work (Osada et al.) that considers adding Gaussian noise for OOD detection, and that empirical results are not directly related to the theoretical ones.

**Reviewer Concerns:**

1) (Outstanding) This concern was clearly not addressed sufficiently, and I believe it remains a fundamental flaw of the paper. The authors' analysis in Section 3 assumes that OOD labels are available, which is not the case in practice. The analysis is superficial, derivative of previous works (Caterini & Loaiza-Ganem), and would be somewhat obvious to experts in the field. Section 4 addresses the problem that OOD labels would not be available in practice, but introduces auxiliary knowledge to effectively determine those labels rather than rely solely on information from the density model. The authors argue in the discussion that prior work (e.g. Kamkari et al.) also uses auxiliary information in the form of LID estimates. However, this argument misses the point that Kamkari et al. extract LID estimates from the density model itself, meaning that they are not auxiliary at all. This is very different from the authors' proposal of SPEM which requires datasets external to the task and training additional feature extractors that can access labeled in-distribution data.

2) (Addressed) The inequality does not always hold, but was brought up in a specific scenario. OOD detection is not solved now. Properties of an optimal embedding model were discussed.

3) (Outstanding) While the theoretical results are correct as written, they do not address the practical reality of the likelihood OOD paradox problem, since they assume entropy of the OOD points can be increased without affecting the entropy of the ID points. In practice, one does not have labels for OOD/ID at test time in order to add noise only to OOD points. The authors' method SPEM and empirical results use external information not related to the likelihood model in order to provide pseudo-labels. Hence this does not show that increasing entropy is a feasible solution to the OOD paradox. Instead, it shows that one can contrive ways of labeling OOD points using feature extractors trained on external data, and with access to some ID labeled data. This is irrelevant to the likelihood OOD paradox problem.

In summary, I strongly agree with the points raised by Reviewer maRj, which I believe are fair, insightful, and correct. The paper is fundamentally flawed and does not address the research question it sets out to solve.

To the authors: Please pay careful attention to the reasoning laid out by Reviewer maRj which should be helpful for improving your work. The research community is interested in understanding why the likelihood-OOD paradox occurs, and how the paradox can be resolved either by manipulating the information encoded within likelihood models themselves (e.g. LID estimates), or by altering the training process of the likelihood models to make them better at OOD detection without sacrificing generative quality. Introducing externally trained feature extractors does not give insight on any of these directions.

**Reviewer Scores:**

Reviewer maRj 2 -> 2. This reviewer had discussions early in the discussion period which were extensive, and mentioned wanting to strongly reject the paper, although they did not change their score out of courtesy.

Reviewer k3Ki 8 -> 8

Reviewer WMtm 2 -> 2 This reviewer did not have a chance to discuss, but their concerns were largely consistent with maRj, and I believe are outstanding.

---

### Decision · Program_Chairs · 2026-01-26

Reject